# A drug repurposing approach reveals targetable epigenetic pathways in *Plasmodium vivax hypnozoites*

Steven P Maher[1]*, Malina A Bakowski[2], Amélie Vantaux[3], Erika L Flannery[4], Chiara Andolina[5], Mohit Gupta[6], Yevgeniya Antonova-Koch[2], Magdalena Argomaniz[7], Monica Cabrera-Mora[8], Brice Campo[9], Alexander T Chao[4], Arnab K Chatterjee[2], Wayne T Cheng[7], Vorada Chuenchob[4], Caitlin A Cooper[1], Karissa Cottier[10], Mary R Galinski[8,11], Anke Harupa-Chung[4], Hana Ji[7], Sean B Joseph[2], Todd Lenz[6], Stefano Lonardi[12], Jessica Matheson[13], Sebastian A Mikolajczak[4], Timothy Moeller[10], Agnes Orban[3], Vivian Padín-Irizarry[1,14], Kastin Pan[2], Julie Péneau[3], Jacques Prudhomme[6], Camille Roesch[3], Anthony Ruberto[1], Saniya S Sabnis[7], Celia L Saney[7], Jetsumon Sattabongkot[15], Saleh Sereshki[12], Sangrawee Suriyakan[5], Ratawan Ubalee[16], Yinsheng Wang[17,18], Praphan Wasisakun[5], Jiekai Yin[18], Jean Popovici[3], Case W McNamara[2], Chester Joyner[7,8], François H Nosten[5,19], Benoît Witkowski[3], Karine G Le Roch[6], Dennis E Kyle[1]*

[1]Center for Tropical and Emerging Global Disease, University of Georgia, Athens, United States; [2]Calibr, a division of The Scripps Research Institute, La Jolla, United States; [3]Malaria Molecular Epidemiology Unit, Institute Pasteur of Cambodia, Phnom Penh, Cambodia; [4]Novartis Institute for Tropical Diseases, Novartis Institutes for Biomedical Research, Emoryville, United States; [5]Shoklo Malaria Research Unit, Mahidol-Oxford Tropical Medicine Research Unit, Mae Sot, Thailand; [6]Department of Molecular, Cell, and Systems Biology, University of California, Riverside, Riverside, United States; [7]Center for Vaccines and Immunology, College of Veterinary Medicine, University of Georgia, Athens, United States; [8]International Center for Malaria Research, Education and Development, Emory Vaccine Center, Emory National Primate Research Center, Emory University, Atlanta, United States; [9]Medicines for Malaria Venture (MMV), Geneva, Switzerland; [10]BioIVT Inc, New York, United States; [11]Division of Infectious Diseases, Department of Medicine, Emory University, Atlanta, United States; [12]Department of Computer Science and Engineering, University of California, Riverside, Riverside, United States; [13]Department of Microbiology and Immunology, University of Otago, Dunedin, New Zealand; [14]School of Sciences, Clayton State University, Morrow, United States; [15]Mahidol Vivax Research Unit, Mahidol University, Bangkok, Thailand; [16]Department of Entomology, Armed Forces Research Institute of Medical Sciences (AFRIMS), Bangkok, Thailand; [17]Department of Chemistry, University of California, Riverside, Riverside, United States; [18]Environmental Toxicology Graduate Program, University of California, Riverside, Riverside, United States; [19]Centre for Tropical Medicine and Global Health, Nuffield Department of Medicine, University of Oxford, Oxford, United Kingdom

*For correspondence:
STEVEN.MAHER@uga.edu
(SPM);
Dennis.Kyle@uga.edu (DEK)

## eLife assessment

This paper reports a large drug repurposing screen based on an in vitro culture platform to identify compounds that can kill Plasmodium hypnozoites. This **valuable** work adds to the current repertoire of anti-hypnozoites agents and uncovers targetable epigenetic pathways to enhance our understanding of this mysterious stage of the Plasmodium life cycle. The data presented here are based on **solid** methodology and represent a starting point for further investigation of epigenetic inhibitors to treat P. vivax infection. This paper will be of interest to Plasmodium researchers and more broadly to readers in the fields of host-pathogen interactions and drug development.

**Abstract** Radical cure of *Plasmodium vivax* malaria must include elimination of quiescent 'hypnozoite' forms in the liver; however, the only FDA-approved treatments are contraindicated in many vulnerable populations. To identify new drugs and drug targets for hypnozoites, we screened the Repurposing, Focused Rescue, and Accelerated Medchem (ReFRAME) library and a collection of epigenetic inhibitors against *P. vivax* liver stages. From both libraries, we identified inhibitors targeting epigenetics pathways as selectively active against *P. vivax* and *P. cynomolgi* hypnozoites. These include DNA methyltransferase inhibitors as well as several inhibitors targeting histone post-translational modifications. Immunofluorescence staining of *Plasmodium* liver forms showed strong nuclear 5-methylcystosine signal, indicating liver stage parasite DNA is methylated. Using bisulfite sequencing, we mapped genomic DNA methylation in sporozoites, revealing DNA methylation signals in most coding genes. We also demonstrated that methylation level in proximal promoter regions as well as in the first exon of the genes may affect, at least partially, gene expression in *P. vivax*. The importance of selective inhibitors targeting epigenetic features on hypnozoites was validated using MMV019721, an acetyl-CoA synthetase inhibitor that affects histone acetylation and was previously reported as active against *P. falciparum* blood stages. In summary, our data indicate that several epigenetic mechanisms are likely modulating hypnozoite formation or persistence and provide an avenue for the discovery and development of improved radical cure antimalarials.

## Introduction

Of the six species of *Plasmodium* that cause malaria in humans (*Ansari et al., 2016*), *Plasmodium vivax* is the most globally widespread (*Howes et al., 2016*). Vivax malaria now accounts for the most malaria episodes in countries with successful falciparum malaria control programs (*Price et al., 2020*). Controlling vivax malaria is complicated by the ability of *P. vivax* sporozoites, the infectious stage inoculated by mosquitoes, to invade hepatocytes and become quiescent (*Wells et al., 2010*; *White et al., 2014*). These quiescent 'hypnozoites' persist, undetectable, for months or even years before resuming growth and initiating a 'relapse' blood stage infection, leading to subsequent transmission back to mosquitoes (*Adams and Mueller, 2017*). New evidence suggests this transmission is expedited and silent as *P. vivax* liver merozoites can immediately form gametocytes instead of first having to establish an asexual stage blood infection, such as is the case for *P. falciparum* (*Roth et al., 2018*; *Adapa et al., 2019*; *Schäfer et al., 2020*; *Mancio-Silva et al., 2022*). Clinically, a compound with radical cure efficacy is one that removes all parasites from the patient, including hypnozoites in the liver (*Campo et al., 2015*).

Hypnozoites are refractory to all antimalarials except the 8-aminoquinolines, which were first identified over 70 years ago using low-throughput screening in avian malaria models (*Rangel and Llinás, 2021*). Primaquine was the first 8-aminoquinoline widely used for radical cure; however, efficacy is contingent on a large total dose administered in a 7- to 14-day regimen, leading to adherence problems and infrequent use in malaria control programs of endemic countries (*Taylor et al., 2019*). Tafenoquine–chloroquine was developed from primaquine as an improved single dose for radical cure (*Llanos-Cuentas et al., 2019*), but a recent clinical trial shows tafenoquine lacks efficacy when co-administered with the common antimalarial dihydroartemisinin-piperaquine, calling into question tafenoquine's suitability in areas of high chloroquine resistance (*Sutanto et al., 2023*). Furthermore, 8-aminoquinolines cannot be administered to pregnant women or glucose-6-phosphate dehydrogenase-deficient individuals and are ineffective in persons with specific cytochrome P450

genotypes (**Baird, 2019**). For these reasons, the discovery and development of new chemical classes with radical cure activity are needed (**Burrows et al., 2017**).

Modern drug discovery typically relies on phenotypic screening and protein target identification (**Schenone et al., 2013**). For malaria, this approach ensures hits are acting on parasite targets and enables rational drug design, leading to several promising novel classes of antimalarials (**Kuhen et al., 2014**; **Forte et al., 2021**). However, due to lower cost and higher feasibility, current high-throughput screening for new antimalarials focuses almost entirely on blood or liver schizonts (**Avery et al., 2014**; **Antonova-Koch et al., 2018**). High-throughput antimalarial screening with a target chemo-profile for killing hypnozoites has only recently been made possible with the introduction of cell-based phenotypic screening platforms featuring a monolayer of hepatocytes infected with sporozoites, a portion of which go on to form hypnozoites (**Valenciano et al., 2022**). While the first hypnozonticidal hits from these platforms are just now being reported (**Maher et al., 2021**), protein target identification approaches for hypnozonticidal drug discovery are in their infancy as the transcriptome of hypnozoites has only recently been reported and robust methods for genetic manipulation of *P. vivax* are still underdeveloped (**Ruberto et al., 2022**; **Bermúdez et al., 2018**).

To address the lack of radical cure drug leads and targets, we used our advanced *P. vivax* liver stage platform to first screen the Repurposing, Focused Rescue, and Accelerated Medchem (ReFRAME) library (**Janes et al., 2018**). This library consists of approximately 12,000 developmental, approved, and discontinued drugs with the expectation that the repurposing of compounds with some optimization or regulatory success could expedite the decade-long path drugs typically progress through from discovery to licensure (**Janes et al., 2018**). To accomplish this screen, we assembled an international collaboration with laboratories in malaria-endemic countries whereby vivax-malaria patient blood was collected and fed to mosquitoes to produce sporozoites for infecting primary human hepatocytes (PHHs) in screening assays performed on-site. Interestingly, two structurally related compounds used for treating hypertension, hydralazine and cadralazine, were found effective at killing hypnozoites. Because these inhibitors have been shown to modulate DNA methylation (**Cornacchia et al., 1988**; **Singh et al., 2009**), we pursued and confirmed the existence of methyl-cytosine modifications in *P. vivax* sporozoite and liver stages. Having found in the ReFRAME screen a class of hits targeting an epigenetic pathway, we decided to confirm the importance of epigenetics in *P. vivax* hypnozoites and screened an additional commercial epigenetic inhibitor library using an improved version of our screening platform. Hypnozoites were found to be susceptible to several classes of epigenetic inhibitors, including six distinct histone deacetylase inhibitors and two inhibitors targeting histone methylation. To further assess the importance of histone acetylation in *P. vivax* liver stages, we tested inhibitors previously reported to be directly acting on *P. falciparum* acetyl-CoA synthetase, thereby modulating the pool of acetyl-CoA available for histone acetylation (**Summers et al., 2022**). We found MMV019721 selectively kills *P. vivax* and *P. cynomolgi* hypnozoites, implicating acetyl-CoA synthetase as an additional hypnozonticidal drug target. This work demonstrates that in lieu of traditional molecular biology methods, our screening platforms identify multiple, druggable epigenetic pathways in hypnozoites and add to the growing body of evidence that epigenetic features underpin biology in *P. vivax* and *P. cynomolgi* sporozoite and liver stages (**Ruberto et al., 2022**; **Dembélé et al., 2014**; **Muller et al., 2019**; **Toenhake et al., 2023**).

## Results
### ReFRAME library screening cascade, hit identification, and confirmation

Chemical biology approaches have shown that hypnozoites become insensitive to most legacy antimalarials after 5 days in culture, indicating they must mature following hepatocyte infection (**Maher et al., 2021**; **Posfai et al., 2020**). Hypnozoite maturation was also noted in recent single-cell transcriptomic analyses of *P. vivax* liver stages, which demonstrate distinct population clusters of maturing and quiescent hypnozoites (**Mancio-Silva et al., 2022**; **Ruberto et al., 2022**). Importantly, discovery and development of hit compounds with radical cure activity in vivo, which includes elimination of hypnozoites in the liver of malaria patients (**Campo et al., 2015**), requires screening against mature hypnozoites in vitro (**Zeeman et al., 2016**). While our 8 day *P. vivax* liver stage platform, in which sporozoites are infected into PHHs and then allowed to mature for 5 days before being treated with test compound (**Maher, 2021**), has been used for screening small libraries against mature hypnozoites

(*Maher et al., 2021*), the size of the ReFRAME library (12,823 compounds tested at 10 µM) presented a logistical challenge. We anticipated that dozens of *P. vivax* cases, each with a unique genetic background, would be needed to produce the sporozoites required to screen the 40 microtiter plates containing the library. To preclude the complex process of regular international shipments of infected mosquitoes, the *P. vivax* liver stage platform was successfully adapted and set up in research labs in two distinct malaria endemic areas, the Shoklo Malaria Research Unit (SMRU) in Thailand and the Institute Pasteur of Cambodia (IPC). The screening library was divided between both sites to enable concurrent progress; ultimately, 36 *P. vivax* cases from either site were needed to complete the primary screen over the course of 18 months (*Figure 1A*, *Figure 1—figure supplement 1*, *Supplementary file 1*).

We selected 72 compounds for confirmation of activity against hypnozoites in a dose–response format. These compounds were counter-screened for additional antimalarial activity against *P. falciparum* blood stages and *P. berghei* liver schizonts and tested for cytotoxicity against HEK293T and HepG2 human cell lines (*Table 1*). Following confirmation in dose–response assays, some hits exhibited moderate selectivity and potency, with $pEC_{50}$'s ranging from 5.42 to 7.07 ($pEC_{50}$ is the inverse log of potency in M concentration, e.g. $pEC_{50}$ 3 = 1 mM, $pEC_{50}$ 6 = 1 µM, and $pEC_{50}$ 9 = 1 nM) (*Table 1*). Colforsin daropate, rhodamine 123, and poziotinib are used to treat cancer and have known human targets, indicating that the targeted host pathways may be critical for hypnozoite persistence. As an example, poziotinib inhibits HER2, a tyrosine protein kinase associated with the downregulation of apoptosis and metastasis (*Kavarthapu et al., 2021*). We recently reported that host apoptotic pathways are downregulated in *P. vivax*-infected hepatocytes (*Ruberto et al., 2022*). Poziotinib could therefore act by upregulating apoptotic pathways in infected host cells. MS-0735, an analog of our previously reported hypnozonticidal hit, MMV018983 (*Maher et al., 2021*), is a

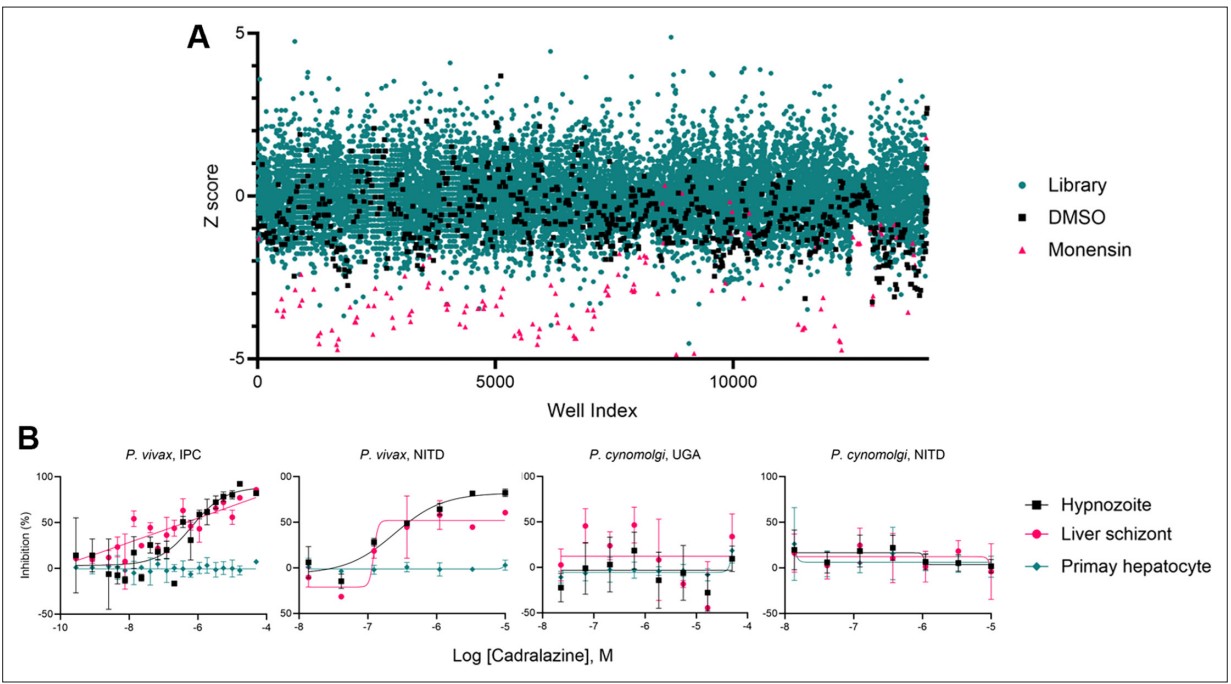

**Figure 1.** Hypnozonticidal hit detection and confirmation. (**A**) Index chart depicting the primary screen of the ReFRAME library against *P. vivax* hypnozoites in an 8-day assay. Hypnozoite counts were normalized by mean quantity per well for each plate (*Z*-score). Teal: library, black: DMSO, red: 1 µM monensin. (**B**) Dose–response curves for cadralazine against *P. vivax* and *P. cynomolgi* liver forms in 8-day assays at the IPC, UGA, and NITD. All replicate wells were plotted together from all independent experiments (*n* = 3 for *P. vivax* at IPC, *n* = 1 for *P. vivax* at NITD, *n* = 2 for *P. cynomolgi* at UGA, and *n* = 4 for *P. cynomolgi* at NITD), bars represent SEM.

The online version of this article includes the following source data and figure supplement(s) for figure 1:

**Figure supplement 1.** ReFRAME screen run detail and hit structures.

**Figure supplement 2.** Select ReFRAME hits confirmed at Novartis Institute for Tropical Diseases (NITD).

**Figure supplement 3.** Pharmacokinetics of cadralazine in nonhuman primates.

**Source data 1.** Source data for *Figure 1* and supporting figures.

**Table 1.** Dose–response confirmation and counterscreens of primary screen hits and analogs.

Primary screen hits and structurally or mechanistically related compounds were tested by dose–response in 8 day *P. vivax* liver stage assays at Institute Pasteur of Cambodia and counterscreened against *P. berghei* liver schizonts, *P. falciparum* asexual blood stages of strain Dd2 and W2, and human cell lines HEK293T and HepG2. Values represent $pEC_{50}$ or $pCC_{50}$ ± SD of all independent experiments ($n$ = 2–6) for which a $pEC_{50}$ or $pCC_{50}$ was obtained. An asterisk (*) indicates only one independent experiment resulted in a calculated $pEC_{50}$ or $pCC_{50}$. $pEC_{50}$ is the inverse log of potency in M concentration, e.g. $pEC_{50}$ 3 = 1 mM, $pEC_{50}$ 6 = 1 µM, and $pEC_{50}$ 9 = 1 nM.

| Compound | Status | P. vivax hypnozoites IPC | P. vivax liver schizonts IPC | Primary human hepatocytes IPC | P. berghei liver schizonts | P. falciparum asexual blood stage, strain Dd2 | Cytotoxicity, HEK293T | Cytotoxicity, HepG2 |
|---|---|---|---|---|---|---|---|---|
| | | ($pEC_{50}$ ± SD) | ($pEC_{50}$ ± SD) | ($pCC_{50}$ ± SD) | ($pEC_{50}$ ± SD) | ($pEC_{50}$ ± SD) | ($pCC_{50}$ ± SD) | ($pCC_{50}$ ± SD) |
| *Antihypertensives* | | | | | | | | |
| Cadralazine | Registered | 6.33 ± 0.29 | 6.33 ± 0.18 | < 5.00 | < 5.00 | < 4.90 | < 4.40 | 4.43* |
| Pildralazine | Discontinued | 6.08 ± 0.27 | ≤ 5.95 | < 5.00 | < 5.00 | < 4.90 | < 4.40 | 4.74* |
| Hydralazine | Registered | 5.75* | 5.42* | < 5.00 | < 5.00 | < 4.90 | < 4.40 | 4.51* |
| Budralazine | Registered | < 5.00 | < 5.00 | < 5.00 | 5.88 ± 0.4 | < 4.90 | < 4.40 | < 4.40 |
| Dihydralazine | Preclinical | < 5.00 | < 5.00 | < 5.00 | 5.53 ± 0.14 | 5.07 ± 0.07 | 4.7 ± 0.06 | 4.50 ± 0.11 |
| Endralazine | Discontinued | < 5.00 | < 5.00 | < 5.00 | < 5.00 | < 4.90 | 4.51* | 4.47* |
| Mopidralazine | Discontinued | < 5.00 | < 5.00 | < 5.00 | < 5.00 | < 4.90 | < 4.40 | < 4.40 |
| Todralazine | Unknown | < 5.00 | < 5.00 | < 5.00 | < 5.00 | < 4.90 | < 4.40 | < 4.40 |
| Dramedilol | Phase I | < 5.00 | < 5.00 | < 5.00 | < 5.00 | < 4.90 | 4.73 ± 0.06 | 4.60 ± 0.06 |
| RGH-5526 | Phase I | < | < 5.00 | < 5.00 | < 5.00 | < 4.90 | 4.87 ± 0.19 | 4.67 ± 0.12 |
| *Anticancer* | | | | | | | | |
| Colforsin daropate | Registered | 7.07* | < 5.00 | < 5.00 | < 5.00 | < 4.90 | 4.71 ± 0.17 | 4.41* |
| Rhodamine 123 | Phase I | 5.23 ± 0.31 | ≤ 5.48 | < 5.00 | < 5.00 | 5.28 ± 0.08 | 5.28 ± 0.3 | 4.65 ± 0.07 |
| PAN-811 | Phase II | < 5.00 | < 5.00 | < 5.00 | 5.91 ± 0.29 | 5.66 ± 0.54 | 6.03 ± 0.23 | 5.77 ± 0.13 |
| Poziotinib | Phase II | < 5.00 | < 5.00 | < 5.00 | 5.23 ± 0.1 | 5.25 ± 0.03 | 5.27 ± 0.22 | 4.72 ± 0.16 |
| *Other* | | | | | | | | |
| Narasin | Animal use | 5.79 ± 0.2 | 6.50* | < 5.00 | 9.09 ± 0.42 | 7.92 ± 0.13 | 7.57 ± 1.07 | 6.66 ± 0.58 |
| MS-0735 | Preclinical | 5.42* | ≤ 5.48 | < 5.00 | 6.22 ± 0.07 | 5.38 ± 0.09 | 6.07 ± 0.22 | 6.05 ± 0.21 |
| Plasmocid | Discontinued | ≤ 5.48 | ≤ 5.95 | < 5.00 | 5.70 ± 0.27 | 6.74 ± 0.56 | 4.96 ± 0.14 | 4.95 ± 0.37 |

The online version of this article includes the following source data for table 1:

**Source data 1.** Source data for *Table 1*.

ribonucleotide-reductase (RNR) inhibitor and used as an antiviral. The apparent need for nonreplicating hypnozoites to produce deoxyribonucleosides for DNA synthesis is peculiar. However, it has been reported that RNR is also critical for DNA damage repair (*Elledge et al., 1992*), is important for maintaining cancer cell dormancy (*Evans and Lin, 2015*), and is expressed in *P. vivax* liver schizonts and hypnozoites (*Ruberto et al., 2022*). We also rediscovered previously reported hypnozonticidal compounds included in the library, including the ionophore narasin (*Maher et al., 2021*) and the 8-aminoquinoline plasmocid (*Schmidt and Schmidt, 1949*; *Figure 1—figure supplement 1*, *Table 1*).

From our analysis of primary screen activity, we noted several hydrazinophthalazine vasodilators were potentially active (*Figure 1—figure supplement 1*) and selected 10 hydrazinophthalazine analogs for dose–response confirmation and counterscreen assays. Three hydrazinophthalazines analogs – cadralazine, pildralazine, and hydralazine – were active against mature hypnozoites, with cadralazine displaying the best combination of potency ($pEC_{50}$ = 6.33 ± 0.33), maximal inhibition near 100%, and selectivity over PHH (>21-fold), HEK293T (>85-fold), and HepG2 (>79-fold) cells (*Figure 1B*, *Table 1*). Hydralazine, which was FDA-approved in 1953, is currently one of the world's most-prescribed antihypertensives, and on the WHO list of essential medicines (*World Health Organization, 2019*). Cadralazine, which was developed in the 1980s as an improvement over hydralazine,

was abandoned due to side effects and only licensed in Italy and Japan (*McTavish et al., 1990*). Hydrazinophthalazines have been shown to inhibit human DNA methyltransferases (DNMT) (*Cornacchia et al., 1988*; *Singh et al., 2009*) and hydralazine has also been recently used to study potential DNA methylation patterns in the *P. falciparum* asexual blood stages (*Ponts et al., 2013*). Similar to our previous report (*Ponts et al., 2013*), these hydrazinophthalazines were inactive when tested against *P. berghei* liver schizonts, *P. cynomolgi* asexual blood stages, and *P. falciparum* asexual blood stages (*Table 1—source data 1*), suggesting that hypnozoite quiescence may be biologically distinct from developing schizonts (*Maher et al., 2021*). While hydrazinophthalazines may act on infected hepatocytes and not directly on the parasite, their distinct selectivity suggests that their effect is likely on a host or parasite pathways and not simply due to cytotoxicity in the host cell. Hydralazine and cadralazine were not identified as potential hits in any of the 112 bioassay screens of the ReFRAME published to date (*Su, 2024*), suggesting these compounds specifically target *P. vivax* liver stages and not promiscuously active compounds.

Methods for the robust culture of *P. vivax* hypnozoites were only recently reported, leading to several new reports on hypnozoite biology and radical cure drug discovery (*Roth et al., 2018*, *Gural et al., 2018*). Consequentially, some hypnozoite-specific discoveries appear to be platform-specific (*Mancio-Silva et al., 2022*; *Ruberto et al., 2022*). Select hits were shared with the Novartis Institute for Tropical Diseases (NITD), where the hypnozonticidal activity and potency of cadralazine ($pEC_{50}$ = 6.09 ± 0.45), hydralazine ($pEC_{50}$ = 6.20), and poziotinib ($pEC_{50}$ = 6.17) were independently confirmed in a similar 8-day *P. vivax* screening platform using a *P. vivax* case from southern Thailand (*Figure 1B*, *Figure 1—figure supplement 2*). Independent confirmation of these hits indicates their activities are not merely platform-specific and are, rather, more broadly descriptive of hypnozoite chemo-sensitivity.

Following our screening and hit confirmation, we investigated the potency, in vivo stability, and tolerability profile of our confirmed hits and chose cadralazine and hydralazine for repurposing as radical cure antimalarials. Currently, the gold-standard model for preclinical assessment of in vivo anti-relapse efficacy is rhesus macaques infected with *Plasmodium cynomolgi*, a zoonotic, relapsing species closely related to *P. vivax* (*Joyner et al., 2015*). Because we found cadralazine substantially more potent against hypnozoites than hydralazine, it was selected for a rhesus macaque pharmacokinetic study in which plasma levels were measured over 24 hr following an oral dose of 1 mg/kg, which was calculated to be well-tolerated, and 30 mg/kg, which was calculated to likely cause drug-induced hypotension (*Hauffe and Dubois, 1984*; *Leonetti et al., 1988*; *Bonardi et al., 1983*). The 30 mg/kg dose resulted in maximum plasma concentration of 13.7 μg/ml (or 48.2 μM) and half-life of 2.19 ± 0.24 hr, which was sufficient to cover the in vitro $EC_{90}$ for several hours without noticeable side effects (*Figure 1—figure supplement 3*). As another prerequisite for in vivo validation, we next sought to confirm and measure the potency of cadralazine and other ReFRAME hits against *P. cynomolgi* B strain hypnozoites in vitro using an 8-day assay featuring primary simian hepatocytes (PSH) at NITD. While poziotinib was active against *P. cynomolgi* hypnozoites when tested in two of three different PSH donor lots ($pEC_{50}$ = 5.67 and 5.95) (*Supplementary file 2*) hydralazine and cadralazine were found inactive when tested in all three different PSH donor lots (*Figure 1B*, *Supplementary file 2*). This negative result was later confirmed in an 8-day, simianized version of the platform at the University of Georgia (UGA) using the *P. cynomolgi* Rossan strain infected into two different PSH lots (*Figure 1B*). Altogether, these data highlight potential differences between *P. vivax* and *P. cynomolgi* and challenge the gold-standard model for preclinical assessment of in vivo anti-relapse efficacy in rhesus macaques.

## Synergy between cadralazine and 5-azacytidine

As molecular tools to validate drug target in *P. vivax* are limited, we further interrogated the possible mechanism of action of hydrazinophthalazines using drug combination studies to assess synergy, additivity, or antagonism (*Summers et al., 2022*). We used 5-azacytidine, a known DNMT inhibitor (*Christman, 2002*), to investigate its effects on cadralazine treatment. When tested alone in dose–response from 50 μM, 5-azacytidine had no effect on hypnozoites. However, when added to cadralazine in fixed ratio combinations ranging from 8:1 to 1:8, 5-azacytidine increased the potency of cadralazine by ~2-fold across several combinations in two independent experiments (*Figure 2*, *Figure 2—figure supplement 1*). The most potent effect was detected using a 2:1 fixed ratio of cadralazine:5-azacytidine, resulting in an equivalent $EC_{50}$ decrease from 470 to 216 nM.

## Immunofluorescent detection of DNA methylation in *P. vivax* and *P. cynomolgi* liver stages

To further investigate if cadralazine could interact with *P. vivax* target(s), we aimed to detect and quantify DNA methylation in the *P. vivax* and *P. cynomolgi* genomes. Previous studies had identified the presence of low-level 5-methylcytosine (5mC), 5-hydroxymethylcytosine (5hmC), and 5hmC-like marks throughout the *P. falciparum* genome (*Ponts et al., 2013*; *Lucky et al., 2023*; *Hammam et al., 2020*; *Lenz et al., 2024*). We first conducted an immunofluorescence staining assay using commercially available anti-5mC and anti-5hmC monoclonal antibodies to identify evidence of DNA methylation in *P. vivax* liver stages at 6 days post-infection. We found clear evidence of 5mC, but not 5hmC, in both schizonts and hypnozoites, morphologically consistent with the presence of 5mC in the parasite's nucleus (*Figure 3A*, *Figure 3—figure supplements 1–3*). To segregate signals coming from the host hepatic nuclei, we used automated high-content imaging analysis on hundreds of individual *P. vivax* liver stage parasites as an unbiased approach for quantifying 5mC signal within parasites. Image masks were generated to quantify the area of 5mC or 5hmC stain within each parasite (*Figure 3—figure supplement 4*). The values were then plotted as stain area per hypnozoite or per schizont (*Figure 3B*). While some evidence of 5hmC-positive forms did appear from this analysis, the net 5hmC area per parasite was found significantly lower when compared to 5mC signals (Kruskal–Wallis tests, for hypnozoites $H(7) = 194.3$, $p < 0.0001$, for schizonts $H(7) = 88.66$, $p < 0.0001$). Similar results on the ratio of 5hmC to 5mC were also recently reported in *P. falciparum* blood stages (*Lenz et al., 2024*), confirming that 5mC marks are the predominant DNA methylation marks in both species.

Given the different susceptibility of *P. cynomolgi* hypnozoites to hydrazinophthalazines as compared to *P. vivax*, we performed automated high-content analysis of 5mC- and 5hmC-stained *P. cynomolgi* M/B-strain liver schizonts and hypnozoites at 8 and 12 days post-infection. Like *P. vivax*, we found both *P. cynomolgi* liver schizonts and hypnozoites are positive for 5mC, but not 5hmC. However, the 5mC stain morphology and intensity were relatively lower in *P. cynomolgi* hypnozoites versus *P. vivax* hypnozoites, suggesting potential divergence of DNA methylation pathways in these two species (*Figure 3—figure supplement 5*).

## Detection of cytosine modifications in *P. vivax* and *P. cynomolgi* sporozoites using liquid chromatography–tandem mass spectrometry and bisulfite sequencing

We next sought to confirm the presence of cytosine methylation in the *P. vivax* and *P. cynomolgi* genomes using mass spectrometry and bisulfite sequencing. We initially assessed that without an available single-cell sequencing approach, sequencing coverage of the parasite's genome would be overwhelmed by the genomic material from the host cell as well as neighboring uninfected hepatocytes

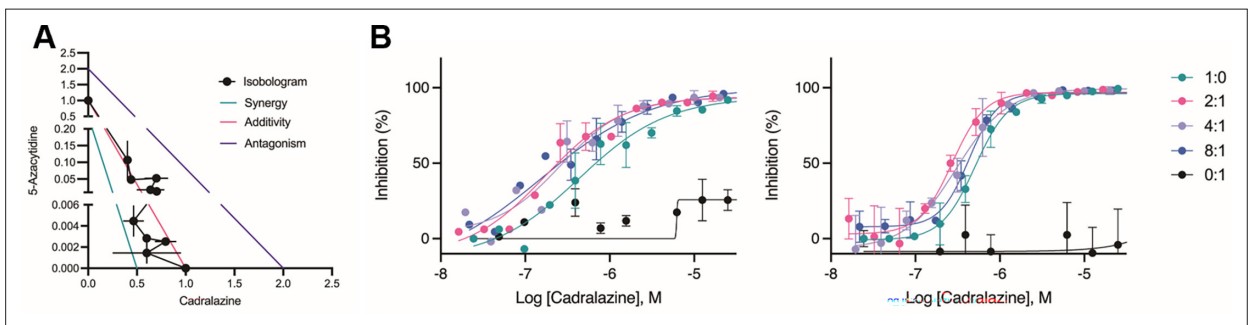

**Figure 2.** Synergistic effect of cadralazine and 5-azacytidine in *P. vivax* liver stage assays. (**A**) Isobologram of cadralazine and 5-azacytidine activity against hypnozoites in fixed ratios of 1:0, 8:1, 6:1, 4:1, 2:1, 1:1, 1:2, 1:4, 1:6, 1:8, and 0:1, bars represent SD of FICs from two independent experiments. (**B**) Dose–response curves for cadralazine at the most synergistic fixed ratios (2:1, 4:1, and 8:1) against hypnozoites. Cadralazine alone is represented as 1:0, 5-azacytidine alone is represented as 0:1 and plotted on the cadralazine chart for comparison. Left and right charts represent two independent experiments, bars represent replicate wells at each dose.

The online version of this article includes the following source data and figure supplement(s) for figure 2:

**Figure supplement 1.** Synergistic effect of cadralazine and 5-azacytidine in *P. vivax* liver stage assays.

**Source data 1.** Source data for *Figure 2* and supporting figures.

(*Ruberto et al., 2022*). We therefore collected sufficient genomic material from *P. vivax* and *P. cynomolgi* sporozoites to analyze the nucleoside mixture arising from the enzymatic digestion of genomic DNA by liquid chromatography–tandem mass spectrometry as well as for detection of DNMT activity using a commercial in vitro DNA methylation assay (*Ponts et al., 2013*). While we detected 5mC and DNMT activity in *Plasmodium*-enriched samples with these approaches, possible contamination by the mosquito's microbiota could not be excluded (*Figure 4—figure supplement 1*). We next analyzed DNA methylation loci at single-nucleotide resolution using bisulfite sequencing of $3 \times 10^7$ *P. vivax* sporozoites, generated from three different cases, as well as $3 \times 10^7$ *P. cynomolgi* sporozoites (*Figure 4A, B*). A total of 161 and 147 million high-quality reads were sequenced for *P. vivax* and *P. cynomolgi* samples, respectively (*Supplementary file 3*). The average 5mC level detected across all cytosines was 0.49% and 0.39% for *P. vivax* and *P. cynomolgi*, respectively. These percentages are comparable to the 0.58% methylation level detected in *P. falciparum* blood stages (*Ponts et al., 2013*), but likely underestimate methylated loci considering the coverage we achieved (see methods).

We then monitored the distribution of detected 5mC along the *P. vivax* and *P. cynomolgi* chromosomes (*Figure 4A*, *Figure 4—figure supplements 2 and 3*) and observed a stable methylation level throughout the genomes, including in telomeric and sub-telomeric regions. We further examined the context of genome-wide methylations and, similar to what we previously observed in *P. falciparum* (*Ponts et al., 2013*), methylation was detected as asymmetrical, with CHH (where H can be any nucleotide but G) at 69.5% and 70.5%, CG at 16% and 15.7%, and CHG at 14.3% and 13.64%, for *P. vivax* and *P. cynomolgi*, respectively (*Figure 4C*). We then measured the proportion of 5mC in the various compartments of gene bodies (exons, the introns, promoters, and terminators) as well as strand specificity (*Figure 4D, E*). We observed a slightly increased distribution of 5mC in promoters and exons compared to the intronic region, as well as in the template versus non-template strand, in *P. vivax* and *P. cynomolgi*. These results were consistent with previous data obtained in *P. falciparum*

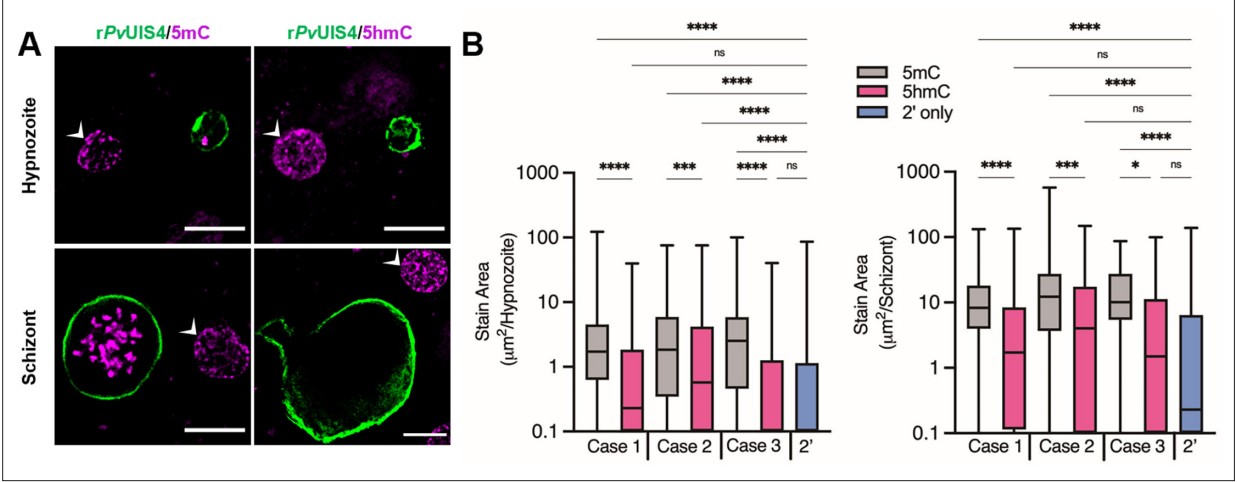

**Figure 3.** Cytosine modifications in *P. vivax* liver forms. (**A**) Immunofluorescent imaging of a 5mC-positive (left) or 5hmC-negative (right) *P. vivax* hypnozoite (top) and schizont (bottom) at day 6 post-infection. White arrows indicate hepatocyte nuclei positive for 5mC or 5hmC. Bars represent 10 μm. (**B**) High-content quantification of 5mC or 5hmC stain area within hypnozoites or schizonts from sporozoites generated from three different *P. vivax* cases. Significance determined using Kruskal–Wallis tests, for hypnozoites $H(7) = 194.3$, $p < 0.0001$, for schizonts $H(7) = 88.66$, $p < 0.0001$, with Dunn's multiple comparisons, $*p < 0.05$, $***p < 0.001$, $****p < 0.0001$, ns = not significant. Line, box, and whiskers represent median, upper and lower quartiles, and minimum-to-maximum values, respectively, of all hypnozoites ($177 \leq n \leq 257$) or all schizonts ($30 \leq n \leq 142$) in culture for each case, 2' indicates a secondary stain only control.

The online version of this article includes the following source data and figure supplement(s) for figure 3:

**Source data 1.** Source data for *Figure 3* and supporting figures.

**Figure supplement 1.** Cytosine modifications in *P. vivax* liver forms, full panels from case 1 (expanded from *Figure 3*).

**Figure supplement 2.** Cytosine modifications in *P. vivax* liver forms, full panels from case 2.

**Figure supplement 3.** Cytosine modifications in *P. vivax* liver forms, full panels from case 3.

**Figure supplement 4.** High-content analysis of cytosine modifications and *P. vivax* liver stage population metrics.

**Figure supplement 5.** Cytosine modifications in *P. cynomolgi* M/B strain liver forms.

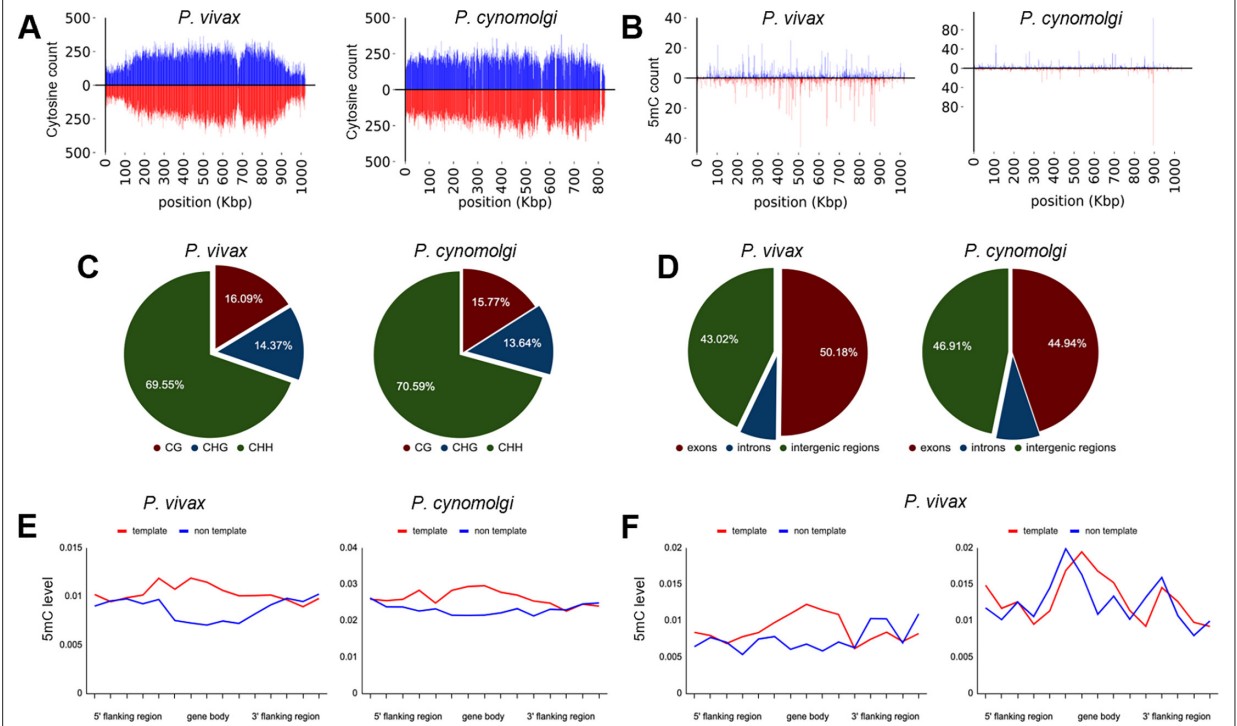

**Figure 4.** Density of cytosine and methylated cytosine (5mC) in sporozoites. (**A**) CG content of chromosome 1 for *P. vivax* and *P. cynomolgi*. The total number of cytosines was quantified on each strand using 1 kbp long non-overlapping windows. (**B**) The total number of methylated cytosines was quantified on each strand using 1 kbp long non-overlapping windows. (**C**) The number of 5mC present in all possible contexts (CG, CHG, and CHH) quantified throughout the genome of *P. vivax* and *P. cynomolgi*. (**D**) Repartitioned 5mC quantity within different compartments of the genome in *P. vivax* and *P. cynomolgi*. (**E**) Strand specificity of 5mC for all genes in *P. vivax* and *P. cynomolgi*. Flanking regions and gene bodies were divided into five bins, and the methylation level of each bin was averaged among all genes. Red: template strand, blue: non-template strand. (**F**) The previously reported mRNA abundance of *P. vivax* sporozoites was retrieved (*Antonova-Koch et al., 2018*) and genes ranked. The 5mC levels in 5′ flanking regions, gene bodies, and 3′ flanking regions were placed into five bins and are shown for highly expressed (90th percentile, left) and weakly expressed (10th percentile, right) genes. Red: template strand, blue: non-template strand.

The online version of this article includes the following figure supplement(s) for figure 4:

**Figure supplement 1.** Measurement of DNA methylation and DNA methyltransferase (DNMT) in *P. vivax* and *P. cynomolgi* sporozoites.

**Figure supplement 2.** Cytosine and methylation density plots for *P. vivax* sporozoites.

**Figure supplement 3.** Cytosine and methylation density plots for *P. cynomolgi* sporozoites.

and in plants (*Ponts et al., 2013*; *Lucky et al., 2023*). Such a strand specificity of DNA methylation patterns can affect the affinity of the RNA polymerase II and impact transcription; thus, we compared methylation levels to previously reported transcriptomic data from *P. vivax* sporozoites (*Muller et al., 2019*). The 5mC levels in 5′ flanking regions, gene bodies, and 3′ flanking regions were placed into five bins and compared to mRNA abundance, revealing an inverse relationship between methylation and mRNA abundance in the proximal promoter regions and the beginning of the gene bodies, with highly expressed genes appearing hypomethylated and weakly expressed genes hypermethylated (*Figure 4F*). These results suggest that methylation level in proximal promoter regions as well as in the first exon of the genes may affect, at least partially, gene expression in malaria parasites. While these data will need to be further validated and linked to hypnozoite formation at a single-cell level, we have determined that 5mC is present at a low level in *P. vivax* and *P. cynomolgi* sporozoites and could control liver stage development and hypnozoite quiescence.

## Assay improvements and epigenetic inhibitor library screen

The success of the original screening platform protocol and secondary confirmation of several of our initial hits provided us an invaluable opportunity to develop an improved radical cure screening assay. The current iterations of our screening platform rely on high-content analysis of parasitophorous

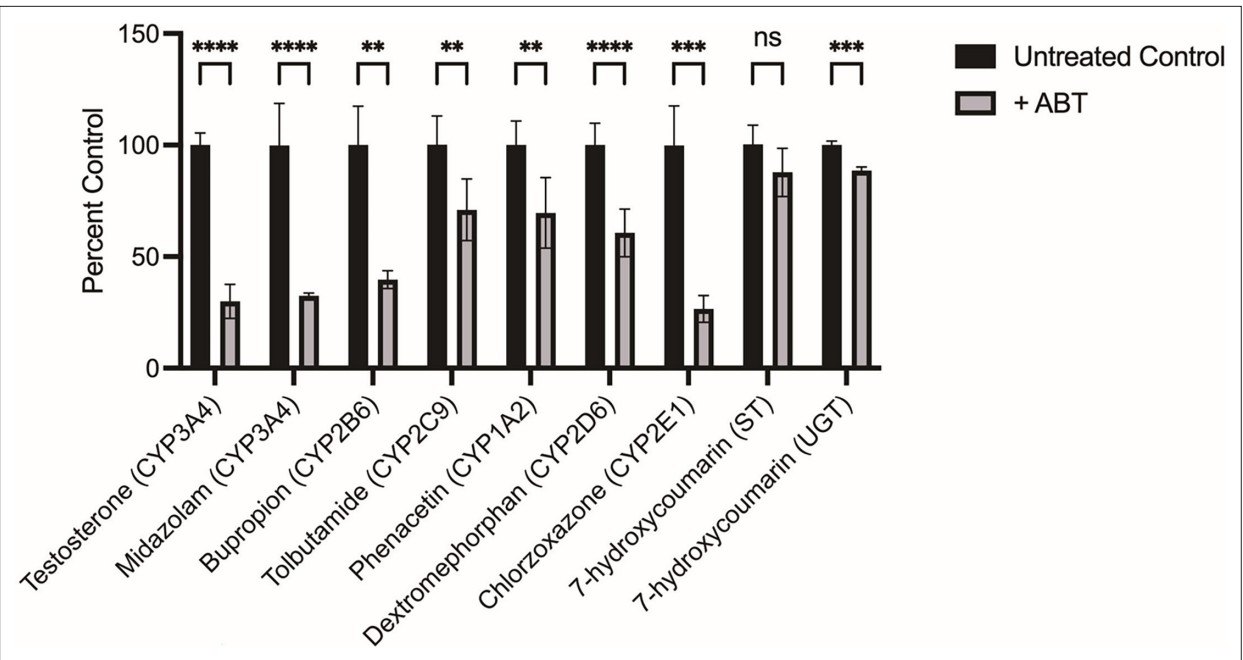

**Figure 5.** Characterization of primary human hepatocyte (PHH) metabolism following 1-aminobenzotriazole (1-ABT) treatment. PHH lot BGW was seeded in 384-well plates and cultured for 7 days before treatment with 100 µM 1-ABT for 1 hr, followed by addition of substrates for 1 hr and collection for analysis by mass spectrometry. Data are combined from two independent experiments, bars represent SD of all replicates. Significance determined by Student's *t* tests, ****p < 0.0001, ***p < 0.001, **p < 0.01, ns, not significant.

The online version of this article includes the following source data and figure supplement(s) for figure 5:

**Figure supplement 1.** Monensin activity in all control wells based on primary human hepatocyte (PHH) lot.

**Figure supplement 2.** ReFRAME hits re-confirmed in a *P. vivax* 12-day 1-aminobenzotriazole (1-ABT) assay.

**Figure supplement 3.** Epigenetic inhibitor library screen and hits.

**Source data 1.** Source data for *Figure 5* and supporting figures.

vacuole staining of the forms that persist up to the assay endpoint (*Roth et al., 2018*; *Schafer et al., 2018*). During the course of the ReFRAME primary screen, we found the day 8 endpoint was sufficient for some hit compounds to act. However, other compounds like the 8-aminoquinolines exhibit a 'delayed death' phenotype, which leads to a false-negative result (*Maher et al., 2021*). We therefore extended the assay by 4 days to allow attenuated forms to be cleared from the culture (*Maher et al., 2021*). Also, as our screening assays were performed with multiple lots of PHH and PSH, we detected some lot-specific results, possibly due to compound instability in the presence of hepatic metabolism (*Figure 5—figure supplement 1*). We therefore tested the metabolism inhibitor 1-aminobenzotriazole (1-ABT) in culture media to minimize the effect of lot-specific hepatic metabolism (*Ortiz de Montellano and Mathews, 1981*). We used a cytochrome P450 functional assay specific to CYP3A4 and determined that 100 µM of 1-ABT was sufficient to completely reduce CYP3A4 activity in both basal and rifampicin-induced PHH (*Figure 5—figure supplement 1*). This effect was further confirmed and quantified by mass spectrometry after 1 hr of treatment at 100 µM 1-ABT. We not only detected a 75% decrease in CYP3A4 activity, but also a more than 60% reduction of CYP2B6 and CYP2E1 activity along with lesser effects on CYP2C9, CYP1A2, and CYP2D6 (*Figure 5*). These changes were incorporated into our original 8-day protocol to design an improved 12-day assay (*Maher, 2021*) that we then validated by re-testing 12 ReFRAME hits. The modified assay did not drastically affect the potency of most hits (*Figure 5—figure supplement 3*), but helped resolve the hypnozonticidal activity of poziotinib (pEC$_{50}$ = 6.05), which had been previously confirmed in *P. vivax* and *P. cynomolgi* assays performed at NITD (*Figure 5—figure supplement 2*, *Supplementary file 2*, *Figure 1—figure supplement 2*). This assay was then used in all follow-up experiments.

To further confirm the importance of epigenetics in hypnozoite biology (*Dembélé et al., 2014*), we obtained a commercially available library containing 773 compounds targeting various inhibitors

**Table 2.** Additional epigenetic inhibitors with activity against *P. vivax* liver stages.

| Epigenetic inhibitor | Target(s) | Hypnozoite pEC$_{50}$ ± SD | Liver schizont pEC$_{50}$ ± SD | PHH nuclei pCC$_{50}$ ± SD |
|---|---|---|---|---|
| Panobinostat | HDAC | 6.98 ± 0.18 | 7.00 ± 0.15 | 5.68 ± 0.18 |
| AR42 | HDAC | 6.11 ± 0.24 | 6.30 ± 0.20 | 5.29 ± 0.27 |
| Raddeanin A | HDAC | 5.95 ± 0.00 | 5.38 ± 0.13 | 5.49 ± 0.02 |
| 666–15 | CREB | 5.88 ± 0.12 | 5.79 ± 0.03 | 5.46 ± 0.03 |
| Abexinostat | HDAC | 5.48 ± 0.00 | 5.26 ± 0.33 | < 5.00 |
| MI2 | Menin-MLL | 5.48 ± 0.00 | 5.48 ± 0.00 | < 5.00 |
| Givinostat | HDAC | 5.35 ± 0.45 | 5.35 ± 0.18 | < 5.00 |
| MMV019721 | *P. falciparum* ACS | 5.31 ± 0.03 | 5.25 ± 0.45 | < 5.00 |
| Cerdulatinib | SYK/JAK | 5.33 ± 0.20 | 5.26 ± 0.31 | < 5.00 |
| Pracinostat | HDAC | 5.32 ± 0.13 | 5.72 ± 0.20 | < 5.00 |
| CCT241736 | FLT3/Aurora Kinase | 5.24 ± 0.33 | 5.24 ± 0.34 | < 5.00 |
| Cyproheptadine | SETD | 5.24 ± 0.34 | 5.46 ± 0.03 | < 5.00 |

HDAC: histone deacetylase. CREB: cAMP response element-binding protein. FLT3: fms-like tyrosine kinase 3. *P. falciparum* ACS: *P. falciparum* acetyl CoA synthetase. SYK: spleen tyrosine kinase. JAK: Janus kinase. SETD: SET domain containing histone lysine methyltransferase. Mean and standard deviation are from two or more independent experiments.

The online version of this article includes the following source data for table 2:

**Source data 1.** Source data for *Table 2*.

of epigenetic enzymes or pathways. These compounds were tested at 10 μM against *P. vivax* liver stages at both SMRU and IPC sites. We confirmed our initial hits in dose–response assays resulting in selective hypnozonticidal potency for 11 compounds targeting five different epigenetic mechanisms (*Table 2*). This includes the histone deacetylase inhibitors panobinostat (pEC$_{50}$ = 6.98 ± 0.18), AR42 (pEC$_{50}$ = 6.11 ± 0.24), abexinostat (pEC$_{50}$ = 5.48 ± 0.00), givinostat (pEC$_{50}$ = 5.35 ± 0.45), practinostat (pEC$_{50}$ = 5.32 ± 0.13), and raddeanin A (pEC$_{50}$ = 5.95 ± 0.00). Histone methyltransferase inhibitor hits included MI2 (pEC$_{50}$ = 5.48 ± 0.00), a compound that targets the interaction between menin (a global regulator of gene expression), and MLL (a DNA-binding protein that methylates histone H3 lysine 4 *Cierpicki and Grembecka, 2014*), and cyproheptadine (pEC$_{50}$ = 5.24 ± 0.34), which targets the SET-domain-containing lysine methyltransferase (*Hirano et al., 2018*). These results corroborate our hypothesis that epigenetic pathways regulate hypnozoites (*Dembélé et al., 2014*; *Muller et al., 2019*). Other hits, including 666-15 (pEC$_{50}$ = 5.88 ± 0.12), an inhibitor of the transcription factor cAMP response element-binding protein (*Xie et al., 2015*), and cerdulatinib (pEC$_{50}$ = 5.33 ± 0.20), a kinase inhibitor, suggest that signaling pathways may also be important for quiescence (*Glennon et al., 2023*).

Having identified several histone deacetylase inhibitors as directly or indirectly active on hypnozoites, we next screened compounds previously reported as inhibitors of *P. falciparum* acetyl-CoA synthetase (ACS), with downstream effects on histone acetylation (*Summers et al., 2022*). We found that one compound, MMV019721, was selectively active on mature *P. vivax* hypnozoites (*Table 2*). Given the evidence, MMV019721 is directly targeting *P. falciparum* ACS (*Summers et al., 2022*), this result suggests ACS also is a hypnozonticidal drug target. While the molecular techniques needed to confirm the direct interaction of MMV019721 and ACS in *P. vivax* are currently underdeveloped, our data supplement recent reports describing epigenetics as important regulators in *P. vivax* and *P. cynomolgi* at different stages of the parasite life cycle (*Ruberto et al., 2022*; *Muller et al., 2019*; *Toenhake et al., 2023*).

.

## Discussion

Herein we demonstrate several significant advances that progress radical cure antimalarial drug discovery and development, including the first report of screening a medium-sized (>10,000) compound library against mature hypnozoites as well as detection of novel hits with mechanisms unrelated to that of 8-aminoquinolines. Identification of these hits was made possible following the establishment of a complex logistical operation in which the sporozoites used for screening were produced by feeding *P. vivax*-infected blood from malaria patient isolates to mosquito colonies at malaria research institutes in two countries in Southeast Asia. Our international collaboration overcame several logistical hurdles to obtain positive Z-factors for most screening plates. Hits were also confirmed via dose–response, indicating that expanded screening directed against *P. vivax* liver stages is likely to produce more hypnozoite-specific hits (*Table 1*).

The only class of FDA-approved compounds for radical cure, the 8-aminoquinolines, was not discovered from in vitro drug screening. Instead, they were discovered using animal models, including the *P. cynomolgi*-infected rhesus macaque system (*Rangel and Llinás, 2021*). The 8-aminoquinolines function through generation of reactive oxygen species affecting both the host and parasite and lack a distinct parasite target (*Dong et al., 2022*; *Watson et al., 2022*; *Camarda et al., 2019*; *Davidson et al., 1981*). As such, this work represents one of the first applications of a radical cure development pipeline to begin with in vitro screening against *P. vivax* hypnozoites and end with attempted confirmation using *P. cynomolgi* radical cure models. While our screen generated positive results against *P. vivax*, we found mixed results against *P. cynomolgi* hypnozoites in vitro (*Figure 1B*, *Supplementary file 2*). While further studies will be needed to confirm that targets of our hits are parasite- or host-directed, our data show there is sufficient diversity in gene expression, structural biology, or mechanisms of hepatic quiescence between *P. cynomolgi* and *P. vivax* hypnozoites that some newly identified hits may be species-specific. While this result could also be attributed to differential metabolism in human and monkey hepatocytes (*Liang et al., 2020*), the rhesus macaque radical cure model is currently considered an important prerequisite for continued drug development, including efficacy testing in controlled human infections. The role of this model in the radical cure drug development cascade may need to be reevaluated as some compounds identified as promising for the radical cure of *P. vivax* may be abandoned too quickly due to the lack of activity against *P. cynomolgi*. This result highlights the need for further development and validation of *P. vivax*-specific animal models (*Flannery et al., 2022*). Furthermore, this report adds to the broader discussion surrounding the successes and challenges of drug repurposing (*Krishnamurthy et al., 2022*). While direct repositioning of a known drug as a safe treatment for a new indication is the ideal outcome, it can serve as advanced starting points for further optimization and still has the potential for reducing the time and cost involved in developing an efficacious therapy.

In addition to the identification of promising new hits and direction, our data suggests that epigenetic control of pathogenic dormancy via DNA methylation is a pathway that could be potentially targeted by future antimalarials. This pathway has already been described for several disease agents capable of dormancy, including cancer cells (*Ferrer et al., 2020*) and tuberculosis (*Shell et al., 2013*). DNA methylation has also been validated as controlling critical processes in plants, which share evolutionary traits with *Plasmodium* (*Merrick, 2021*). DNA methylation in the genus *Plasmodium* was first described in *P. falciparum* blood stages (*Ponts et al., 2013*) and has been associated with gene expression, transcriptional elongation, and parasite growth (*Lucky et al., 2023*; *Hammam et al., 2021*; *Lenz et al., 2024*). Previous experiments have shown that hydralazine can directly inhibit DNA methylation in nuclear extracts of blood stage parasites but also inhibit a recombinant functional fragment of the *P. falciparum* DNMT (*Ponts et al., 2013*). We pursued several biomolecular approaches to confirm that cadralazine may also interact with *P. vivax* DNMT in liver stage parasites. Due to technical limitations, we used a two-drug combination study in which the known DNMT inhibitor 5-azacytidine potentiated cadralazine against *P. vivax* hypnozoites (*Figure 2*). While we continue to develop new protocols and confirm the direct interaction of cadralazine with *P. vivax*, we successfully confirmed 5mC marks in *P. vivax* and *P. cynomolgi* liver stage parasites using both immunofluorescence and whole genome bisulfite sequencing assays (*Figures 3 and 4*).

The current model of hypnozoite quiescence suggests RNA-binding proteins (RBPs) drive hypnozoite formation by preventing translation of target mRNAs associated with schizogony (*Toenhake et al., 2023*). In this model, histone acetylation results in euchromatin at the loci of RBPs, resulting in

their expression and ongoing quiescence. Hypothetically, HDAC inhibitors would favor quiescence, while a treatment that decreases histone acetylation would favor schizogony. This model somewhat contrasts with our present findings that HDAC inhibitors and the ACS inhibitor MMV019721 successfully kill hypnozoites in vitro (*Table 2*). It is, however, likely that the identified RBPs are part of broader gene networks which, when perturbed by sudden modulation of epigenetic features such as DNA methylation and histone acetylation, result in a lethal level of dysregulation. While we still need to develop *P. vivax* transgenic lines to successfully study hypnozoite biology and further validate potential drug targets (*Voorberg-van der Wel et al., 2020*; *Wel et al., 2021*), the chemical probes that we described in this report could be used in combination with single-cell technology to more precisely perturb hypnozoites and refine our understanding of epigenetic pathways regulating hypnozoite formation and survival.

## Materials and methods
### ReFRAME library description and plating
The ReFRAME library was curated by assembling a list of developmental and FDA-approved chemistry from three databases (GVK Excelra GoStar, Clarivate Integrity, and Citeline Pharmaprojects). The original library consisted of 36 384-well plates (ReF01-ReF36, *Supplementary file 1*) containing 11,871 test compounds (*Janes et al., 2018*). While the original library was being screened, an additional set of four 384-well plates (ReF38–ReF41, *Supplementary file 1*) was added to the library, totaling 12,823 test compounds (*Su, 2024*). Source plates were made from the master library at Calibr at Scripps Research such that 3–5 µl of 10 mM solution was added to each well of a sterile, conical-bottom 384-well plate (Greiner Bio-One cat 784261). Most compounds were diluted in DMSO; however, a subset was diluted in water due to limited DMSO solubility. Plates were sealed and shipped on dry ice to SMRU and IPC and stored at –20°C prior to use. Column 24 of each plate was filled with 5 µl DMSO to serve as negative control wells. Control compounds included 1 mM monensin (positive control for hypnozoite and schizont activity), 1 mM the phosphatidylinositol 4-kinase inhibitor (PI4Ki) KDU691 or MMV390048 (positive control for schizont activity), 1 mM atovaquone (negative control for radical cure activity) and 10 mM tafenoquine (clinically relevant control for hypnozoite activity) (*Roth et al., 2018*; *Maher et al., 2021*).

### Ethical approval for human subjects and animal use
The Thai human subjects protocols for this study were approved by the Institutional Ethics Committee of the Thai Ministry of Public Health and the Oxford Tropical Medicine Ethical Committee (TMEC 14-016 and OxTREC 40-14). The Cambodian human subjects protocols for this study were approved by the Cambodian National Ethics Committee for Health Research (100NECHR, 104NHECR, 111NECHR, 113NHECR, and 237NHECR). Protocols conformed to the Helsinki Declaration on Ethical Principles for Medical Research Involving Human Subjects (*World medical association general assembly, 2004*) and informed written consent was obtained for all volunteers or legal guardians. *P. cynomolgi* sporozoites were generated at Emory National Primate Research Center (ENPRC) using procedures approved by the Emory University Institutional Animal Care and Use Committee (PROTO201900110), as well as at UGA using procedures approved by UGA's Institutional Animal Care and Use Committee (A2020 03-002-Y3-A15). *P. cynomolgi* sporozoites were also produced at the Armed Forces Research Institute of Medical Science under an IACUC-approved animal use protocol in an AAALAC International-accredited facility with a Public Health Services Animal Welfare Assurance and in compliance with the Animal Welfare Act and other federal statutes and regulations relating to laboratory animals (22-10). *P. berghei* sporozoites were generated by the Sporocore at UGA using procedures approved by UGA's Institutional Animal Care and Use Committee (A2016 06-010-Y1-A0 and A2020 01-013-Y2-A3). Pharmacokinetic studies were conducted at WuXi AppTec Co, Ltd, in accordance with the WuXi IACUC standard animal procedures along with the IACUC guidelines that are in compliance with the Animal Welfare Act (*National research council committee, 2011*).

### ReFRAME primary screen against *P. vivax* liver stages
The complete, step-by-step protocol for the *P. vivax* liver stage assay is published (*Maher, 2021*). In summary, 2 days after assay plates (Greiner Bio-One cat 781956) were seeded with PHH, sporozoites

were dissected from mosquito salivary glands and allowed to infect cultures. The ReFRAME library was screened using the original, 8-day radical cure assay, in which developing liver schizonts and mature, PI4Ki-insensitive hypnozoites were treated on days 5–7 post-infection (*Roth et al., 2018*; *Maher et al., 2021*). On treatment days, a pintool was used to transfer 40 nl of compounds from the source plates into 40 µl of media in the assay plates, resulting in a 1000-fold dilution of all compounds. A single PHH lot, UBV, was first used for screening; however, once all available cryovials were used, screening was completed with lot BGW (*Supplementary file 1*). Screening was initiated at SMRU until a second screening site was established at IPC, where all unfinished source plates were shipped and assayed. Some plates were assayed more than once in order to obtain a single run with a sufficient *Z'* factor of >0.0 or two moderate-quality runs allowing for identification of reproducibly active wells (*Supplementary file 1*). Quantification of parasite growth was performed by fixing and staining cultures with recombinant mouse-anti *P. vivax* Upregulated in Infectious Sporozoites 4 (r*Pv*UIS4) (*Schafer et al., 2018*), followed by high-content imaging and analysis using an ImageXpress Micro (Molecular Devices) or Lionheart FX (Agilent). Hypnozoites were classified as forms of less than 125 µm² growth area.

## Normalization, hit selection, and dose–response confirmation in *P. vivax* liver stage assays

Primary screening data were imported into Genedata Screener, Version 15.0.1-Standard and normalized to DMSO (neutral) and inhibitor (monensin) control-treated wells (neutral controls minus inhibitors). For four plates where the monensin control failed due to solubility issues combined with PHH lot variability (*Figure 5—figure supplement 1*), data were normalized using the Robust *Z*-score method, which calculates for each well the Robust *Z*-score (number of standard deviations off the median) based on the statistics of the compound wells per plate. Genedata multiplicative pattern correction was applied to adjust for plate edge effects. Sixty-two most active (≥67% normalized inhibition of hypnozoite numbers) and non-toxic (≤40% host cell toxicity) compounds and 10 hydrazinophthalazines were selected for reconfirmation in an 8-point 1:3 dose response following the 8 day protocol with PHH lot BGW using a dose–response of monensin and nigericin as redundant positive controls. Once hydralazine and cadralazine were identified as reconfirmed hits, commercially available batches of powder were obtained (budralazine, Chemcruz cat sc-504334 batch D3019, cadralazine, Chemcruz cat sc-500641 batch B2417, and hydralazine, Selleckchem cat s2562 batch S256202) and used for additional reconfirmation runs using the same 8-day protocol (*Figure 1B*, *Table 1*).

## Hit confirmation in *P. cynomolgi* liver stage assays at UGA

*P. cynomolgi* assays at UGA were performed using the step-by-step protocol for the *P. vivax* liver stage assay (*Maher, 2021*) with a few modifications. A Japanese macaque (*Macaca fuscata*) was intravenously infected with *P. cynomolgi* Rossan strain cryopreserved ring stage parasites (*Collins et al., 2009*) and allowed to reach patency. When parasitemia reached approximately 5000 parasites per µl, *An. dirus* mosquitoes were fed directly on the infected animal over a period of 3–4 days. The blood-fed mosquitoes were then checked for infection 6–8 days by dissecting and staining midguts with 2% mercurochrome to detect oocysts. Two experiments were performed, one with PSH lot CWP, and one with PSH lot NPI. Two days after assay plates (Greiner Bio-One cat 781956) were seeded with 20,000 live PSH per well, sporozoites were dissected from mosquito salivary glands at day 16 post-bloodmeal and allowed to infect cultures. Hits were confirmed using the same 8-day radical cure assay. On treatment days, a pin tool was used to transfer 40 nl of compounds from the source plates to the assay plates. Quantification of *P. cynomolgi* liver stage growth was performed by fixing and staining cultures with 100 ng/ml mouse monoclonal antibody 13.3 (anti-GAPDH) obtained from The European Malaria Reagent Repository (http://www.malariaresearch.eu) followed by high-content imaging and analysis using an ImageXpress Micro (Molecular Devices). Hypnozoites were classified as forms of less than 105 µm² growth area.

## Hit confirmation in *P. cynomolgi* and *P. vivax* liver stage assays at NITD

Lots of both PSH and PHH were obtained from BioIVT. Hepatocytes were seeded at 22,000 cells per well in a 384-well plate (Corning cat 356667). Prior to and during the infection, the hepatocytes were cultured in BioIVT CP Medium (cat Z99029) with the addition of 1% penicillin–streptomycin–neomycin

(PSN) mix (Gibco cat 15640055) and 0.1% gentamicin in the case of *P. vivax*. Two days post-seeding, the hepatocytes were infected with sporozoites dissected from the salivary glands of *An. dirus* mosquitoes. Sporozoites were collected in RPMI 1640 (KD Medical cat CUS-0645). Hepatocytes were infected with 10,000 sporozoites per well and spun for 5 min at 200 × g. Once the sporozoites were removed after 24 hr of incubation, the culture media was exchanged to include 5% PSN in the case of *P. cynomolgi*. On days 4, 5, 6, and 7 post-infection, the hepatocytes received fresh compound addition in media. The cells were fixed on day 8 using 4% paraformaldehyde.

Liver stage parasites were detected by immunofluorescence assay. Hepatocytes were permeabilized for 1 hr at room temperature in blocking buffer consisting of 2% bovine serum albumin (Millipore Sigma cat A2153) and 0.2% Triton X-100 (Millipore Sigma cat 648466) in 1× PBS (Gibco cat 20012-027). For *P. cynomolgi* staining, the two in-house primary antibodies used were mouse anti-*Pc*UIS4 monoclonal at 10 ng/ml, and rabbit anti-*Pc*HSP70 polyclonal at 200 ng/ml. For *P. vivax* staining, rabbit anti-*Pv*MIF was used at 1:1000 (*Mikolajczak et al., 2015*). The primary antibodies were diluted in blocking buffer and incubated overnight at 4°C. Hepatocytes were washed thrice with 1× PBS and then incubated with secondary antibodies (Invitrogen cat A11013, RRID:AB_2534080 and A11036, RRID:AB_10563566) used at a 1:1000 dilution and Hoechst 33342 (Invitrogen cat H3570) used at 2 µg/ml for 2 hr at room temperature. After the incubation, the hepatocytes were washed 3 times with 1× PBS and were stored in 50 µl per well of 1× PBS prior to imaging on an ImageXpress Micro (Molecular Devices).

## Confirmed hit counterscreens: *P. falciparum* asexual blood stage at Calibr

The SYBR Green I-based parasite proliferation assay (*Plouffe et al., 2016*) was used to determine the activity of compounds against the asexual blood stage of *P. falciparum* strain Dd2-HLH, a transgenic line expressing firefly luciferase (*Ekland et al., 2011*). Briefly, acoustic compound transfer (Labcyte Echo 550) was used to prepare assay-ready plates to which parasites in assay medium were added and incubated with compounds for 72 hr. SYBR Green I in lysis buffer was used as detection reagent. Fluorescence signal was read on the PHERAstar FSX plate reader (BMG Labtech). Compounds were tested in technical triplicates on different assay plates across three biological replicates performed on different days. Data were uploaded to Genedata Screener, Version 16.0.3-Standard and normalized to DMSO (neutral) and inhibitor control-treated wells (neutral controls minus inhibitors), with 1.25 µM dihydroartemisinin used as a positive control. Dose curves (13 point, 1:3 dilution series) were fitted with the four parameter Hill Equation.

## Confirmed hit counterscreens: *P. falciparum* asexual blood stage at UGA

Budralazine, cadralazine, and hydralazine (same catalog and batches as above) were tested using the [³H]-hypoxanthine drug susceptibility assay as previously described, with some modifications (*Hott et al., 2015*). Strain W2 (*Oduola et al., 1988*; *Canfield et al., 1995*) was grown in continuous culture using RPMI 1640 media containing 10% heat-inactivated type A+human plasma, sodium bicarbonate (2.4 g/l), HEPES (5.94 g/l), and 4% washed human type A+ erythrocytes. Cultures were gassed with a 90% $N_2$, 5% $O_2$, and 5% $CO_2$ mixture and incubated at 37°C. Cultures were sorbitol synchronized to achieve >70% ring stage parasites (*Lambros and Vanderberg, 1979*). Assays were started by establishing a 0.5–0.7% parasitemia and 1.5% hematocrit in complete media. Assays were performed in 96-well plates with a volume of 90 µl/well of parasitized erythrocytes and 10 µl/well of 10× test compound. Dihydroartemisinin was plated as a positive control and DMSO as a negative control. Assay plates were incubated in the above-mentioned gas mixture at 37°C for 48 hr; then, ³H-hypoxanthine (185 MBq, PerkinElmer cat NET177005MC) was added, and plates were incubated for another 24 hr. After 72 hr of incubation, the assay plates were frozen at −80°C. Plates were allowed to thaw at room temperature before well contents were collected onto filtermats using a plate harvester (PerkinElmer). A Micro Beta liquid scintillation counter (PerkinElmer) was used to quantify radiation (counts-per-minute) representing relative parasite growth. Values were normalized to controls and plotted using CDD Vault. Potency values represent means of at least two independent experiments.

## Confirmed hit counterscreens: *P. cynomolgi* asexual blood stage at UGA

Budralazine, cadralazine, and hydralazine (same catalog and batches as above) were tested against *P. cynomolgi* DC strain using the [$^3$H]-hypoxanthine drug susceptibility assay as previously described, with some modifications (*Hott et al., 2015*). *P. cynomolgi* was grown in continuous culture using RPMI 1640 +GlutaMAX media containing 20% heat-inactivated rhesus serum, hypoxanthine (32 mg/l), HEPES (7.15 g/l), additional glucose (2 g/l), and 5% washed rhesus erythrocytes. Cultures were incubated at 37°C under mixed gas conditions of 90% $N_2$, 5% $O_2$, and 5% $CO_2$. Schizonts were synchronized over a 60/20 Percoll gradient to achieve >90% late-stage parasites. Assays were started the following day when ring-stage parasites were present. Parasites were prepped for assay by establishing 0.5% ring-stage parasitemia and 2% hematocrit in complete media without hypoxanthine. Assays were performed in 96-well plates with a volume of 90 µl/well of parasitized erythrocytes and 10 µl/well of 10× test compounds. Compounds were plated from a starting concentration of 5 µM in an 11-point 1:2 dilution series and tested in duplicate. Uninfected RBCs were plated as a positive control, and DMSO was used as a negative control. $^3$H-hypoxanthine (185 MBq, PerkinElmer cat NET177005MC) was then added to all wells and plates were incubated under the previously mentioned conditions for 72 hr. After 72 hr the assay plates were frozen at –80°C. Plates were thawed the following day at room temperature and well contents were collected onto filtermats using a plate harvester (PerkinElmer). A Micro Beta liquid scintillation counter (PerkinElmer) was used to quantify radiation (counts-per-minute) representing relative parasite growth. Values were normalized to controls and plotted using CDD Vault. Potency values represent means of at least two independent experiments.

## Confirmed hit counterscreens: *P. berghei* liver stage at Calibr

For *P. berghei* liver stage assays, a colony of *An. stephensi* mosquitoes was maintained in the UGA Sporocore using methods previously described (*Pathak et al., 2022*). In summary, adults were fed 5% dextrose (wt/vol) and 0.05% para-aminobenzoic acid (wt/vol) soaked into cotton pads and kept at a temperature of 27°C, relative humidity of 75–85%, and a 12 hr light/dark cycle. PbGFP-LUC$_{CON}$ sporozoites were produced as previously described (*Pathak et al., 2022*). In summary, female C57BL/6 or Hsd:ICR(CD-1) mice (Envigo) were injected intraperitoneally with $5 \times 10^6$ to $5 \times 10^7$ blood stage parasites in 500 µl PBS 3–4 days before mosquito infections. Once parasitemia reached 2–6%, mice were anesthetized with 0.5 ml 1.25% 2,2,2-Tribromoethanol (vol/vol, Avertin, Sigma-Aldrich) and placed on top of cage of *An. stephensi* mosquitoes (3–7 days post-emergence) for 20 min to serve as an infectious bloodmeal. Infected mosquitoes were shipped to Calibr, where sporozoites were dissected out of mosquito salivary glands and used for luciferase-based infection assay as previously described (*Swann et al., 2016*). Briefly, HepG2 cells (ATCC cat HB-8065, RRID:CVCL_0027) were infected with freshly dissected sporozoites. The infected cells were incubated with compounds of interest in 1536-well plates for 48 hr, and intracellular parasite growth was measured using bioluminescence. Compounds were tested in technical triplicates on different assay plates across three biological replicates performed on different days. Data were uploaded to Genedata Screener, Version 16.0.3-Standard and normalized to DMSO (neutral) and inhibitor control-treated wells (neutral controls minus inhibitors), with 1 µM KAF156 used as a positive control. Dose curves (13 point, 1:3 dilution series) were fitted with the four parameter Hill Equation.

## Confirmed hit counterscreens: mammalian cell cytotoxicity at Calibr

HepG2 (ATCC cat HB-8065, RRID:CVCL_0027) and HEK293T (ATCC cat CRL-3216, RRID:CVCL_0063) mammalian cell lines were maintained in Dulbecco's modified Eagle medium (DMEM, Gibco) with 10% heat-inactivated HyClone FBS (GE Healthcare Life Sciences), 100 IU penicillin, and 100 µg/ml streptomycin (Gibco) at 37°C with 5% $CO_2$ in a humidified tissue culture incubator. Cultures were routinely confirmed free of mycoplasma via Mycoalert (Lonza) using the manufacturer's protocol. To assay mammalian toxicity of hit compounds, 750 HepG2 and 375 HEK293T cells/well were seeded, respectively, in assay media (DMEM, 2% FBS, 100 U/ml penicillin, and 100 µg/ml streptomycin) in 1536-well, white, tissue culture-treated, solid bottom plates (Corning cat 9006BC) that contained acoustically transferred compounds in a threefold serial dilution starting at 40 µM. After a 72-hr incubation, 2 µl of 50% Cell-Titer Glo (Promega cat G7573) diluted in water was added to the cells and luminescence measured on an EnVision Plate Reader (PerkinElmer).

## Combination drug studies in *P. vivax* liver stages

Powders of cadralazine (same batch as above), 5-azacytidine (Caymen Chem, cat 11164), and nigericin were diluted to 50 mM, 50 mM, and 200 µM, respectively, in DMSO, before being diluted to 100 µM, 100 µM, and 400 nM, respectively, in hepatocyte culture media (BioIVT, cat Z99029). Cadralazine and 5-azacytidine were then plated in the first column of two 96-well plates at volumetric ratios of 1:0, 8:1, 6:1, 4:1, 2:1, 1:1, 1:2, 1:4, 1:6, 1:8, and 0:1 such that the net volume per well was 200 µl (nigericin and DMSO controls were also diluted as such). Each mixture was then diluted in a 12-point, twofold dilution series by mixing 100 µl of mixture to 100 µl media in subsequent columns using a multichannel pipettor. A 384-well *P. vivax* liver stage assay plate was started using the 12-day protocol as above, and on day 5, 6, and 7 post-infection, media was removed from the 384-well plate using the inverted spin method *Maher, 2021* followed by addition of 20 µl of fresh media. Then, a multichannel pipettor was used to transfer 20 µl of the mixtures (made fresh daily) from the 96-well dilution series plates to the 384-well plates, thereby establishing a highest 1:0 and 0:1 treatment dose of 50 µM. The assay was fixed, stained, imaged, and parasite growth quantified as described above. Parasite growth data were normalized to the DMSO control and loaded into Prism (GraphPad) for curve fitting using the setting 'log(inhibitor) vs. response – variable slope (four parameters) least squares fit'. The $EC_{50}$'s of each ratio were used to calculate Fractional Inhibitory Concentrations (FICs) and plot isobolograms as previously described (*Ohrt et al., 2002*).

## Immunofluorescent staining of methyl-cytosine modifications in *P. vivax* liver stages

Sporozoites from three different *P. vivax* cases were infected into PHH lot BGW at day 2 post-seed (for case 1) or day 3 post-seed (for cases 2 and 3) in 384-well plates (Greiner Bio-One cat 781956) using the same methods for initiating *P. vivax* liver stage screening assays described above. Cultures were fixed at day 6 post-infection and stained with rPvUIS4 and Hoechst 33342 as previously described (*Maher, 2021*; *Schafer et al., 2018*). Cultures were then stained with either rabbit anti-5mC monoclonal antibody (clone RM231, Thermo Fisher Scientific cat MA5-24694, RRID:AB_2665309) or rabbit anti-5hmC monoclonal antibody (clone RM236, Thermo Fisher Scientific, cat MA5-24695, RRID:AB_2665308) using methods adapted from those previously described by *Hammam et al., 2020*. In summary, cultures were re-permeabilized with 0.1% (vol/vol) Triton X-100 for 20 min at room temperature and then washed thrice with 1× PBS. Chromatin was then denatured with 4 N HCl for 30 min at room temperature and washed thrice with 1× PBS. The denaturing reaction was then neutralized with 100 mM Tris (pH 8.0) for 10 min at room temperature and washed thrice with 1× PBS. Cultures were then quenched with 50 mM $NH_4Cl$ for 10 min at room temperature and washed thrice with 1× PBS. Cultures were then blocked with 0.1% (vol/vol) Tween 20 and 2% (wt/vol) bovine serum albumin for 10 min at room temperature and washed thrice with PBS. Cultures were then stained with either antibody diluted to 10 µg/ml in PBS overnight at 4°C and washed thrice with 1× PBS. Cultures were then stained with 10 µg/ml Texas Red-conjugated, goat anti-rabbit IgG secondary antibody (Thermo Fisher Scientific, cat T-2767, RRID:AB_2556776) overnight at 4°C and washed thrice with 1× PBS. For a negative stain control, a separate set of infected wells was prepared as above and stained with secondary antibody only (2' control, *Figure 3B*). High-resolution images of individual parasites and PHH nuclei were obtained by capturing eight planes in the Z dimension using a 100× objective on Deltavision Core (GE Healthcare Life Sciences) and deconvoluted using softWoRx (GE Healthcare Life Sciences) (*Figure 3A*, *Figure 3—figure supplements 1–3*). An ImageXpress Micro high-content imager was used to quantify methyl-cytosine modifications for the entire population of parasites from each case. A 20× objective was used to capture 25 fields of view from each well (covering the entire growth area) of the 384-well plate. Using the associated MetaXpress high-content analysis software, the rPvUIS4 stain from each parasite was used to define parasite objects, and the 5mC or 5hmC staining of host cell nuclei was used to define positive methyl-cytosine modification objects. The two-dimensional area of intersection of both objects was then quantified for each parasite, and forms less than 125 µm² were quantified as hypnozoites (*Figure 3—figure supplement 4*).

## Immunofluorescent staining of methyl-cytosine modifications in *P. cynomolgi* liver stages

Japanese macaques (*M. fuscata*) were intravenously infected with *P. cynomolgi* M/B strain (*Joyner et al., 2019*) and allowed to reach patency before skin feeding to *An. dirus* mosquitoes as described above. One round of macaque infection, mosquito dissection, and culture infection was performed with PSH lot NPI, and a second round was performed with PSH lot NNF. Two days after assay plates (Greiner Bio-One cat 781956) were seeded with 20,000 PSH per well, sporozoites were dissected from mosquito salivary glands at day 16 post-bloodmeal and allowed to infect cultures. Cultures were fixed on day 8 (experiment 1) or 12 (experiment 2) post-infection and stained for 5mC and 5hmC as described above. An ImageXpress Micro high-content imager was used to quantify methyl-cytosine modifications for the entire population of *P. cynomolgi* liver stage parasites. A 20× objective was used to capture 25 fields of view from each well (covering the entire growth area) of the 384-well plate. Using the associated MetaXpress high-content analysis software, the GAPDH stain from each liver stage parasite was used to define parasite objects, and the 5mC or 5hmC staining of host cell nuclei was used to define positive methyl-cytosine modification objects. The two-dimensional area of intersection of both objects was then quantified for each parasite, and forms less than 105 µm$^2$ were categorized as hypnozoites (*Figure 3—figure supplement 5*).

## Collection of *P. vivax* and *P. cynomolgi* sporozoites for methyl-cytosine characterization

For quantification of 5mC modification levels by mass spectrometry, sporozoites from 3 different *P. vivax* cases, numbering $18.7 \times 10^6$ from case 1, $101 \times 10^6$ from case 2, and $14.7 \times 10^6$ from case 3, were dissected from infected *An. dirus* mosquitoes at IPC as previously described (*Maher, 2021*) and cryopreserved as previously described (*Singh et al., 2016*). For quantification of DNMT activity from nuclear extracts, sporozoites from two different *P. vivax* cases, numbering $21 \times 10^6$ from case 1 and $20 \times 10^6$ from case 2, were similarly dissected and cryopreserved. To serve as a negative control, salivary glands from uninfected mosquitoes at IPC were similarly dissected and cryopreserved. A total of $4.8 \times 10^6$ sporozoites for mass spec and $34.1 \times 10^6$ sporozoites for DNMT activity assays were also collected from *An. dirus* mosquitoes infected from feeding on a rhesus macaque infected with *P. cynomolgi* M/B strain at ENPRC and cryopreserved as described above. To serve as a negative control, salivary glands and ovaries from uninfected mosquitoes at ENPRC were similarly dissected and cryopreserved. For mapping of methyl-cytosine modifications by bisulfite sequencing, sporozoites from three different *P. vivax* cases, numbering $9.8 \times 10^6$ from case 1, $12.3 \times 10^6$ from case 2, and $15.1 \times 10^6$ from case 3, were dissected from infected *An. dirus* mosquitoes at IPC and cryopreserved as described above. A total of $5.3 \times 10^6$ sporozoites were also collected from *An. dirus* mosquitoes infected from feeding on a rhesus macaque infected with *P. cynomolgi* M/B strain at ENPRC and cryopreserved as described above. Frozen sporozoites and salivary glands were shipped from IPC and ENPRC to University of California, Riverside on dry ice.

## Quantification of 5mC, 5hmC, and 2'-deoxyguanosine (dG) in genomic DNA by LC–MS/MS/MS

Parasite pellets were lysed with 100 µl lysis buffer (20 mM Tris, pH 8.1, 20 mM EDTA, 400 mM NaCl, 1% SDS and 20 mg/ml proteinase K) and incubated at 55°C overnight. Saturated solution of NaCl (0.5× volume of reaction mixture) was subsequently added to the digestion mixture and incubated at 55°C for another 15 min. The samples were centrifuged at 14,500 RCF for 30 min at 4°C and the supernatant was removed to a 1.5-ml microcentrifuge. Genomic DNA (gDNA) was then precipitated with 2× volume of 100% chilled ethanol and resuspended in 95 µl water. Samples were then treated with 3 µl of 10 mg/ml RNase A and 2 µl of 25 units/µl RNase T1 and incubated overnight at 37°C. gDNA was then extracted by chloroform/isoamyl alcohol solution, precipitated again with 100% chilled ethanol, and washed with 70% ethanol. The gDNA pellets were then dissolved in nuclease-free water. One µg of gDNA was enzymatically digested into mononucleosides using nuclease P1 and alkaline phosphatase. Enzymes in the digestion mixture were removed by chloroform extraction. The resulting aqueous layer was dried by using a SpeedVac, and the dried residues were subsequently reconstituted in doubly distilled water. Approximately 5 ng of the DNA digestion mixture was injected for LC–MS/MS/MS analyses for quantifications of 5mC, 5hmC, and dG. An LTQ XL linear ion-trap mass

spectrometer equipped with a nano electrospray ionization source and coupled with an EASY-nLC II system (Thermo Fisher Scientific) was used for the LC–MS/MS/MS experiments. The amounts of 5mC, 5hmC, and dG (in moles) in the nucleoside mixtures were calculated from area ratios of peaks found in the selected-ion chromatograms for the analytes over their corresponding isotope-labeled standards, the amounts of the labeled standards added (in moles), and the calibration curves. The final levels of 5mC and 5hmC, in terms of percentages of dG, were calculated by comparing the moles of 5mC and 5hmC relative to those of dG.

## Extraction of nuclear protein

Cryopreserved sporozoites, or parasites extracted from red blood cells by saponin lysis, were resuspended in 1 ml of cytoplasmic lysis buffer (20 mM HEPES pH 7.9, 10 mM KCl, 1 mM EDTA, 1 mM EGTA, 1 mM dithiothreitol (DTT), 0.5 mM AEBSF, 0.65% Igepal, 1× Roche complete protease inhibitor cocktail) and incubated for 10 min on ice. Nuclei were separated from cytoplasmic fraction by 10 min of centrifugation at 1500 RCF followed by two washes with cytoplasmic lysis buffer and one time wash with ice cold 1× PBS. Nuclei pellets were resuspended in 100 µl of nuclei lysis buffer (20 mM HEPES pH 7.9, 0.1 M NaCl, 1 mM EDTA, 1 mM EGTA, 1 mM DTT, 25% glycerol, 0.5 mM AEBSF, 1× Roche complete protease inhibitor cocktail) for 20 min at 4°C with rotation. Nuclear extracts were cleared by 10 min of centrifugation at 6000 RCF. Protein concentration of nuclear extract was quantified by BCA assay and DNMT assays were performed immediately after estimation of protein concentration.

## DNMT assay

DNMT activity of nuclear extracts from *P. cynomolgi* sporozoites, *P. vivax* sporozoites, and uninfected mosquito salivary glands was measured using the Epiquik DNMT activity/inhibition assay ultra-kit (cat P-3010) following the manufacturer's instructions. Purified bacterial DNMT enzyme was used as a positive control. A blank control was used to subtract the residual background values. Each reaction was performed in duplicate. DNMT activity was measured in relative unit fluorescence per h per mg of protein for 10 min at 1-min intervals.

## Bisulfite conversion and library preparation

*P. cynomolgi* and *P. vivax* sporozoites were lysed using 100 µl of lysis buffer containing 20 mM Tris (pH 8.1), 20 mM EDTA, 400 mM NaCl, 1% SDS (wt/vol) for 30 min at room temperature followed by addition of 20 µl of proteinase K (20 mg/ml) to the pellet and incubated at 55°C overnight. The gDNA mixture was purified with phenol–chloroform followed by chloroform. Precipitation of gDNA was performed using chilled ethanol and treated with RNase A followed by another round of ethanol precipitation. 50 ng of unmethylated lambda DNA was added as a control to each sample before bisulfite conversion of the DNA. 500 ng of gDNA of each sample was used for the bisulfite conversion following the manufacturer's instructions (Epitect fast bisulfite conversion kit, QIAGEN cat 59824). Libraries from bisulfite-converted DNA were prepared using the Accel-NGS methyl-Seq DNA library kit (Swift Biosciences cat 30024). Libraries were generated following the manufacturer's instructions and DNA was cleaned through SPRI select beads (Beckman Coulter). Libraries were sequenced using the NOVASeq platform.

## DNA methylation analysis

Four sets of reads for *P. vivax* and *P. cynomolgi* were analyzed. Read qualities were checked with FastQC v0.11.8. FastQC indicated the presence of adapter contamination and overrepresented k-mers. As a result, (1) the first 9–14 base pairs were trimmed and (2) reads with overrepresented k-mers were discarded (see *Supplementary file 3* for summary statistics after the cleaning step). Reads were mapped against the corresponding reference genomes downloaded from PlasmoDB (namely, PlasmoDB-48_Pfalciparum3D7, PlasmoDB-48_PcynomolgiB, and PlasmoDB-48_PvivaxP01) using Bismark v0.22.2 with default parameters. To determine the bisulfite conversion rate, reads were also mapped against the lambda phage (see *Supplementary file 3* for the conversion rate). Alignment files for the replicates were merged together using Samtools v1.9. Read methylation levels were obtained using Bismark v0.22.2 with default parameters (see *Supplementary file 3*).

A cytosine in the genome was considered methylated if (1) the number of reads covering that cytosine was higher than a given threshold (10 for *P. falciparum*, 5 for *P. vivax*, and 3 for *P. cynomolgi*) and

(2) the ratio of methylated reads over all reads covering a cytosine was higher than a given threshold (we chose 0.1 for this second threshold). Genome-wide cytosine density and methylated cytosine density in *Figure 4A, B* were calculated in 1 kbp non-overlapping sliding windows using a custom script (available at https://github.com/salehsereshki/pyMalaria copy archived at *Sereshki, 2021*). The distribution of CG, CHG, and CHH methylation in *Figure 4C, D* was obtained by computing the number of methylated cytosines in each context over all the methylated cytosines. For the methylation analyses in genes in *Figure 4E*, (1) 500 bp flanking regions and gene body were split into five bins and (2) methylation levels were averaged across all the genes using a custom script (available at the https://github.com/salehsereshki/pyMalaria; *Sereshki, 2021*). To study the correlation between cytosine methylation and gene expression, the same gene body computation was done for the 10% high and low expressed genes using a previously reported *P. vivax* transcriptome (*Muller et al., 2019*). These plots are represented in *Figure 4F*.

## Assessment of effect of 1-ABT on hepatic cytochrome P450 3A4 activity

Two experiments were performed, one on uninduced PHHs, and another on rifampicin-induced PHHs (BioIVT, lot BGW). Cells were thawed and 18,000 live cells/well were seeded into collagen-coated 384-well plates as described above. Media was exchanged every other day until day 7 post-seed when media exchange included a dilution series of 1-ABT. One hour after addition of 1-ABT, cytochrome P450 3A4 activity (CYP3A4) was measured using a luciferin-IPA kit (Promega cat V9001) following the lytic protocol with 3 μM IPA. Lysed well contents were transferred to a white 384-well luminometer plate (Greiner Bio-One cat 201106) before reading on a Spectramax i3X (Molecular Devices) with a 1-s integration time. In the second experiment, cells were similarly seeded and cultured before addition of 25 μM rifampicin (MP Biomedial cat BP2679-250), or an equivalent vol/vol DMSO vehicle control, in media on days 4 and 6. At day 7 post-seed, CYP3A4 activity was measured following addition of 1-ABT as above. The fold change was calculated between induced and uninduced wells at each 1-ABT dilution point.

## Assessment of effect of 1-ABT on hepatic metabolism using mass spectrometry

PHHs (lot BGW, BioIVT) were thawed and 18,000 live cells/well were seeded into collagen-coated 384-well plates as described above. Media was exchanged every other day until day 7 post-seed when cells were treated with 100 μM 1-ABT, or an equivalent vol/vol vehicle control, in media for 1 hr. Cells were then incubated with standard substrates for characterization of phase I and II hepatic metabolism, including: 30 μM 7-hydroxycoumarin (UGT/ST), 40 μM coumarin (CYP2A6), 500 μM chlorzoxazone (CYP2E1), 50 μM dextromethorphan (CYP2D6), 24 μM midazolam (CYP3A4/5), 500 μM S-mephenytoin (CYP2C19), 600 μM testosterone (CYP3A4), 1 mM tolbutamide (CYP2C9), 500 μM phenacetin (CYP1A2), or 400 μM bupropion (CYP2B6). The reaction was stopped at 1 hr by addition of an equal volume of ice-cold methanol. Metabolite formation was quantified using UPLC–MS/MS or LC–MS/MS (7-HC, 7-HCS, and 7-HCG). Samples were thawed, vortexed, and centrifuged for 5 min at 5000 rpm. Standards, controls, blanks, and study samples were added to an HPLC autosampler vial and injected into the UPLC–MS/MS or LC–MS/MS systems. Analyses were run using an Acquity UPLC (Waters) or Agilent 1100 HPLC (Agilent) and Quattro premier XE (Waters) or Quattro Premier ZSpray (Waters) mass spectrometers. Quantification was performed using a quadratic least squares regression algorithm with $1/X^2$ weighting, based on the peak area ratio of substrate or metabolite to its internal standard. Metabolite formation rate was calculated as pmol/min/$10^6$ cells.

## Additional ReFRAME hit confirmation using an improved *P. vivax* liver stage assay

Twelve hits were re-confirmed using the 12-day radical cure assay, implementing three assay improvements *Maher, 2021*. First, 100 μM 1-ABT (Caymen Chem cat 15252) was added to media on treatment days to reduce hepatic metabolism. Second, the assay endpoint was extended 4 days to allow for nonviable liver stage forms to be cleared from cultures and therefore not be quantified during high-content imaging. Third, nigericin replaced monensin as the positive ionophore control. Confirmation

was performed with one independent experiment for all compounds except cadralazine, which was confirmed in four independent experiments.

## Epigenetic inhibitor library screen against *P. vivax* liver stages

The Epigenetic Inhibitor library (Targetmol, cat L1200), containing 773 compounds at 10 mM, was purchased and re-plated in pintool-ready 384-well source plates with 200 µM nigericin and DMSO control wells. The library was screened using the 12-day radical cure assay noted above. The 24 hits exhibiting the highest inhibition against hypnozoites were replated in a dose–response for confirmation of activity in a 12-day radical cure assay as described above. Confirmation was performed with two independent experiments. The ACS inhibitors MMV019721 and MMV084978 were kindly provided by MMV and tested in dose–response in a 12-day radical cure assay as described above. Potency was determined from four independent experiments.

## Acknowledgements

We thank the malaria patients of Thailand and Cambodia for participation in this study. We thank the Sporocore at UGA for generating P. berghei-infected mosquitoes. We are grateful to Calibr's Compound Management and High Throughput Screening Groups for their assistance with this project. HCI data from drug studies was produced by the Biomedical Microscopy Core at UGA, supported by the Georgia Research Alliance. SMRU is part of the Mahidol Oxford Research Unit, supported by the Wellcome Trust of Great Britain (#220211). Material has been reviewed by the Walter Reed Army Institute of Research. There is no objection to its presentation and/or publication. The opinions or assertions contained herein are the private views of the author, and are not to be construed as official or as reflecting true views of the Department of the Army or the Department of Defense. This publication includes data generated at the University of California, San Diego IGM Genomics Center utilizing an Illumina NovaSeq 6000 that was purchased with funding from a National Institutes of Health SIG grant (#S10 OD026929). Funding support was provided by the Bill & Melinda Gates Foundation (#OPP1107194 to Calibr, INV-031788 to CJJ, and #OPP1023601 to DEK), Medicines for Malaria Venture (RD/17/0042 and RD/15/0022 to BW and AV and RD/15/0022 to SPM and DEK), the National Institutes of Allergy and Infectious Diseases of the National Institutes of Health (#HHSN272201200031C to MRG and #1R01 AI136511 to KGLR), and the University of California, Riverside (#NIFA-Hatch-225935 to KGLR).

## Additional information

### Competing interests

Erika L Flannery, Anke Harupa-Chung, Sebastian A Mikolajczak: AH-C, VC, ELF, and SAM are employees of the Novartis Institute for Tropical Disease. Brice Campo: BC is an employee of MMV. Vorada Chuenchob: AH-C, VC, ELF, and SAM are employees of the Novartis Institute for Tropical Disease,. Karissa Cottier, Timothy Moeller: TM and KC are employees of BioIVT. The other authors declare that no competing interests exist.

### Funding

| Funder | Grant reference number | Author |
|---|---|---|
| Bill and Melinda Gates Foundation | #OPP1107194 | Malina A Bakowski<br>Case W McNamara |
| Bill and Melinda Gates Foundation | INV-031788 | Chester Joyner |
| Bill and Melinda Gates Foundation | #OPP1023601 | Dennis E Kyle |
| Medicines for Malaria Venture | RD/17/0042 | Amélie Vantaux<br>Benoît Witkowski |

| Funder | Grant reference number | Author |
|---|---|---|
| Medicines for Malaria Venture | RD/15/0022 | Steven P Maher<br>Amélie Vantaux<br>Benoît Witkowski<br>Dennis E Kyle |
| National Institutes of Health | #HHSN272201200031C | Mary R Galinski |
| National Institutes of Health | 1R01 AI136511 | Karine G Le Roch |
| University of California, Riverside | #NIFA-Hatch-225935 | Karine G Le Roch |

The funders had no role in study design, data collection, and interpretation, or the decision to submit the work for publication. For the purpose of Open Access, the authors have applied a CC BY public copyright license to any Author Accepted Manuscript version arising from this submission

## Author contributions

Steven P Maher, Conceptualization, Resources, Data curation, Formal analysis, Supervision, Funding acquisition, Validation, Investigation, Visualization, Methodology, Writing – original draft, Project administration, Writing – review and editing; Malina A Bakowski, Conceptualization, Resources, Data curation, Formal analysis, Validation, Investigation, Visualization, Methodology, Writing – original draft, Writing – review and editing; Amélie Vantaux, Conceptualization, Resources, Supervision, Funding acquisition, Investigation, Methodology, Project administration, Writing – review and editing; Erika L Flannery, Data curation, Supervision, Validation, Investigation, Writing – review and editing; Chiara Andolina, Magdalena Argomaniz, Monica Cabrera-Mora, Wayne T Cheng, Hana Ji, Saniya S Sabnis, Celia L Saney, Jetsumon Sattabongkot, Ratawan Ubalee, Praphan Wasisakun, Resources; Mohit Gupta, Mary R Galinski, Supervision, Investigation; Yevgeniya Antonova-Koch, Vorada Chuenchob, Sean B Joseph, Todd Lenz, Stefano Lonardi, Timothy Moeller, Agnes Orban, Kastin Pan, Julie Péneau, Jacques Prudhomme, Saleh Sereshki, Sangrawee Suriyakan, Yinsheng Wang, Jiekai Yin, Investigation; Brice Campo, Arnab K Chatterjee, Sebastian A Mikolajczak, Supervision; Alexander T Chao, Anke Harupa-Chung, Data curation, Investigation; Caitlin A Cooper, Resources, Investigation, Methodology, Writing – review and editing; Karissa Cottier, Investigation, Writing – review and editing; Jessica Matheson, Investigation, Visualization; Vivian Padín-Irizarry, Investigation, Methodology, Writing – review and editing; Camille Roesch, Resources, Investigation, Visualization, Methodology, Writing – review and editing; Anthony Ruberto, Formal analysis; Jean Popovici, Resources, Supervision; Case W McNamara, Conceptualization; Chester Joyner, Resources, Supervision, Funding acquisition, Writing – review and editing; François H Nosten, Writing – review and editing; Benoît Witkowski, Resources, Supervision, Funding acquisition, Project administration; Karine G Le Roch, Conceptualization, Data curation, Formal analysis, Supervision, Funding acquisition, Validation, Investigation, Visualization, Methodology, Writing – original draft, Writing – review and editing; Dennis E Kyle, Supervision, Funding acquisition, Project administration, Writing – review and editing

## Author ORCIDs

Steven P Maher https://orcid.org/0000-0002-9560-5656
Malina A Bakowski http://orcid.org/0000-0002-3337-6528
Amélie Vantaux https://orcid.org/0000-0002-7945-961X
Erika L Flannery https://orcid.org/0000-0003-0665-7954
Kastin Pan https://orcid.org/0009-0003-5838-1694
Jacques Prudhomme https://orcid.org/0000-0002-6161-5194
Anthony Ruberto http://orcid.org/0000-0002-3215-9484
Jetsumon Sattabongkot https://orcid.org/0000-0002-3938-4588
Chester Joyner https://orcid.org/0000-0003-1367-2829
François H Nosten https://orcid.org/0000-0002-7951-0745
Dennis E Kyle https://orcid.org/0000-0002-0238-965X

### Ethics

The Thai human subjects protocols for this study were approved by the Institutional Ethics Committee of the Thai Ministry of Public Health and the Oxford Tropical Medicine Ethical Committee (TMEC 14-016 and OxTREC 40-14). The Cambodian human subjects protocols for this study were approved by the Cambodian National Ethics Committee for Health Research (100NECHR, 104NHECR, 111NECHR, 113NHECR, and 237NHECR). Protocols conformed to the Helsinki Declaration on Ethical Principles for Medical Research Involving Human Subjects and informed written consent was obtained for all volunteers or legal guardians.

P. cynomolgi sporozoites were generated at Emory National Primate Research Center (ENPRC) using procedures approved by the Emory University Institutional Animal Care and Use Committee (PROTO201900110), as well as at UGA using procedures approved by UGA's Institutional Animal Care and Use Committee (A2020 03-002-Y3-A15). P. cynomolgi sporozoites were also produced at the Armed Forces Research Institute of Medical Science under an IACUC-approved animal use protocol in an AAALAC International-accredited facility with a Public Health Services Animal Welfare Assurance and in compliance with the Animal Welfare Act and other federal statutes and regulations relating to laboratory animals (22-10). P. berghei sporozoites were generated by the Sporocore at UGA using procedures approved by UGA's Institutional Animal Care and Use Committee (A2016 06-010-Y1-A0 and A2020 01-013-Y2-A3). Pharmacokinetic studies were conducted at WuXi AppTec Co, Ltd, in accordance with the WuXi IACUC standard animal procedures along with the IACUC guidelines that are in compliance with the Animal Welfare Act.

Reviewer #1 (Public Review): https://doi.org/10.7554/eLife.98221.2.sa1
Reviewer #2 (Public Review): https://doi.org/10.7554/eLife.98221.2.sa2
Reviewer #3 (Public Review): https://doi.org/10.7554/eLife.98221.2.sa3
Author response https://doi.org/10.7554/eLife.98221.2.sa4

# Additional files

### Supplementary files

MDAR checklist

Supplementary file 1. Summary of ReFRAME plate (40 plates labeled 1–41, with 37 skipped) run metrics including average hypnozoites and schizont counts per well, $Z'$ factor for 1 µM monensin wells, screening location (Shoklo Malaria Research Unit, Thailand, or Pasteur Institute of Cambodia) primary human hepatocyte (PHH) lot used, and P. vivax patient isolate used. Due to an error during library plating, some plates contained only 1 well of monensin, preventing calculation of a $Z'$ factor for those plates (listed as N.A.).

Supplementary file 2. Potency data ($pEC_{50}$) for select ReFRAME hits against P. cynomolgi liver forms assayed at NITD in primary simian hepatocyte (PSH) lots NDO, NPI, XXJ infected with one batch of P. cynomolgi sporozoites. Cytotoxicity ($pCC_{50}$) against PSH was measured using nuclei counts. Maduramicin is a positive control with activity against P. cynomolgi hypnozoites.

Supplementary file 3. Summary statistics of read sets, percentage of mapped reads, read methylation levels, conversion rate, and genome-wide methylation levels from bisulfite sequencing.

Supplementary file 4. Pharmacokinetic data report from Wuxi for cadralazine in Rhesus macaques.

Supplementary file 5. Contents of the Targetmol Epigenetic Library.

Supplementary file 6. Usage of reagents for experiments and replication.

### Data availability

All bisulfite sequencing data generated in this study can be found in the Sequence Read Archive (SRA) at the NCBI National Library of Medicine (https://www.ncbi.nlm.nih.gov/sra) under the BioProject code PRJNA925570.

The following dataset was generated:

| Author(s) | Year | Dataset title | Dataset URL | Database and Identifier |
|---|---|---|---|---|
| Gupta M, Lenz T, Prudhomme J, Le Roch KG | 2024 | A Drug Repurposing Approach Reveals Targetable Epigenetic Pathways in Plasmodium vivax Hypnozoites | https://www.ncbi.nlm.nih.gov/bioproject/PRJNA925570/ | NCBI BioProject, PRJNA925570 |

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

## Appendix 1

### Immunofluorescent staining of 5mC and 5hmC in *P. vivax* blood stage parasites

An immunofluorescent staining approach has been used to detect both 5mC and 5hmC in *P. falciparum* blood stage parasites (*Lucky et al., 2023*), thus we sought to confirm these marks in *P. vivax* blood stages. *P. vivax* blood samples were collected between 2017 and 2019 by active and passive case detection from individuals residing in Mondulkiri, Eastern Cambodia. The presence of *P. vivax* was determined using an RDT (CareStartTM Malaria Pf/pan RDTs, Accessbio) or microscopy, and monoinfections were confirmed by RT-PCR using species-specific primers (*Canier et al., 2013*). Venous blood used was collected in lithium heparin tubes and immediately processed on-site in a mobile laboratory. Leukocytes were depleted using NWF filters (*Li et al., 2017*). The leukocyte-depleted parasitized red blood cells were cryopreserved using glycerolyte 57 solution (Baxter) and immediately stored in liquid nitrogen (*Russell et al., 2011*). Blood isolates were thawed by addition of 12%, then 1.6%, and then 0.9% (wt/vol) NaCl solution followed by heparin treatment for 10 min at 37°C. Blood stage parasites were then purified from thawed isolates using a KCl-Percoll density gradient (*Rangel et al., 2018*) followed by a wash with RPMI and two washes with 1× PBS. Parasites were then fixed with 3% (vol/vol) paraformaldehyde and 0.01% (vol/vol) glutaraldehyde in 1× PBS for 1 hr at 4°C. After fixation, blood stage parasites were permeabilized, denatured, neutralized, quenched, and blocked as described above. Staining for 5mC and 5hmC was carried out as described above except the primary and secondary antibodies were diluted to 1 µg/ml instead of 10 µg/ml. Parasites were stained with 10 µg/ml Hoechst 33342 for 30 min at room temperature and then washed twice with 1× PBS after staining. Parasites were mounted on a coverslip and imaged with a 100× objective on a Leica DM250. While we did detect 5mC and 5hmC methylation in residual human white blood cells, we could not confirm positive 5mC or 5hmC staining in *P. vivax* blood stage parasites from these isolates (*Appendix 1—figure 1*). These negative results could be due to one or more factors. First, while *P. falciparum* blood stage cultures can reach parasitemias above 10%, *P. vivax* blood stages cannot be propagated in vitro, and the parasitemia of isolates is typically just above the level of detection. Second, *P. vivax* blood stage isolates were cryopreserved before staining, and the stability of DNA methylation after cryopreservation is unknown. Third, the hydrochloric acid treatment needed to denature chromatin during the stain protocol causes red cells to aggregate, thereby making finding and imaging *P. vivax* blood stages difficult.

#### Appendix 1—key resources table

| Reagent type (species) or resource | Designation | Source or reference | Identifiers | Additional information |
|---|---|---|---|---|
| Cell line (*Homo sapiens*, female) | HEK293T | ATCC | ATCC cat:CRL-3216; RRID:CVCL_0063 | Transformed fetal cells |
| Cell line (*Homo sapiens*, male) | HepG2 | ATCC | ATCC cat:HB-8065; RRID:CVCL_0027 | Hepatoblastoma |
| Biological sample (*Homo sapiens*, male) | Primary human hepatocytes | BioIVT | Lot:UBV | Cryopreserved cryoplateable |
| Biological sample (*Homo sapiens*, male) | Primary human hepatocytes | BioIVT | Lot:BGW | Cryopreserved cryoplateable |
| Biological sample (*Homo sapiens*, male) | Primary human hepatocytes, female | BioIVT | Lot:QWK | Cryopreserved cryoplateable |
| Biological sample (*Macaca fascicularis*, male) | Primary simian hepatocytes | BioIVT | Lot:CWP | Cryopreserved cryoplateable |
| Biological sample (*Macaca fascicularis*, male) | Primary simian hepatocytes | BioIVT | Lot:NPI | Cryopreserved cryoplateable |
| Biological sample (*Macaca fascicularis*, male) | Primary simian hepatocytes | BioIVT | Lot:NDO | Cryopreserved cryoplateable |
| Biological sample (*Macaca mulatta*, male) | Primary simian hepatocytes | BioIVT | Lot:XXJ | Cryopreserved cryoplateable |

*Appendix 1 Continued on next page*

*Appendix 1 Continued*

| Reagent type (species) or resource | Designation | Source or reference | Identifiers | Additional information |
|---|---|---|---|---|
| Biological sample (*Macaca mulatta*, male) | Primary simian hepatocytes | BioIVT | Lot:NNF | Cryopreserved cryoplateable |
| Biological sample (*An. dirus*) | Mosquitoes | Shoklo Malaria Research Unit | | Colony maintained on site |
| Biological sample (*An. dirus*) | Mosquitoes | Institute Pasteur of Cambodia | | Colony maintained on site |
| Biological sample (*An. dirus*) | Mosquitoes | Armed Forces Research Institute of Medical Sciences | | Colony maintained on site |
| Biological sample (*An. dirus*) | Mosquitoes | University of Georgia | | Colony maintained on site |
| Biological sample (*Anopheles stephensi*) | Mosquitoes | University of Georgia | | Colony maintained and infected at UGA, shipped to Calibr for *P. berghei* assays |
| Biological sample (*Plasmodium vivax*) | Patient isolate | Shoklo Malaria Research Unit | PID:402389 | Fresh isolate fed to *An. dirus* mosquitoes |
| Biological sample (*Plasmodium vivax*) | Patient isolate | Shoklo Malaria Research Unit | PID:423955 | Fresh isolate fed to *An. dirus* mosquitoes |
| Biological sample (*Plasmodium vivax*) | Patient isolate | Shoklo Malaria Research Unit | PID:425583 | Fresh isolate fed to *An. dirus* mosquitoes |
| Biological sample (*Plasmodium vivax*) | Patient isolate | Shoklo Malaria Research Unit | PID:432054 | Fresh isolate fed to *An. dirus* mosquitoes |
| Biological sample (*Plasmodium vivax*) | Patient isolate | Shoklo Malaria Research Unit | PID:2020-013 | Fresh isolate fed to *An. dirus* mosquitoes |
| Biological sample (*Plasmodium vivax*) | Patient isolate | Shoklo Malaria Research Unit | PID:2020-014 | Fresh isolate fed to *An. dirus* mosquitoes |
| Biological sample (*Plasmodium vivax*) | Patient isolate | Institute Pasteur of Cambodia | PID:Pv593 | Fresh isolate fed to *An. dirus* mosquitoes |
| Biological sample (*Plasmodium vivax*) | Patient isolate | Institute Pasteur of Cambodia | PID:Pv595 | Fresh isolate fed to *An. dirus* mosquitoes |
| Biological sample (*Plasmodium vivax*) | Patient isolate | Institute Pasteur of Cambodia | PID:Pv602 | Fresh isolate fed to *An. dirus* mosquitoes |
| Biological sample (*Plasmodium vivax*) | Patient isolate | Institute Pasteur of Cambodia | PID:Pv603 | Fresh isolate fed to *An. dirus* mosquitoes |
| Biological sample (*Plasmodium vivax*) | Patient isolate | Institute Pasteur of Cambodia | PID:Pv606 | Fresh isolate fed to *An. dirus* mosquitoes |
| Biological sample (*Plasmodium vivax*) | Patient isolate | Institute Pasteur of Cambodia | PID:Pv608 | Fresh isolate fed to *An. dirus* mosquitoes |
| Biological sample (*Plasmodium vivax*) | Patient isolate | Institute Pasteur of Cambodia | PID:Pv609 | Fresh isolate fed to *An. dirus* mosquitoes |
| Biological sample (*Plasmodium vivax*) | Patient isolate | Institute Pasteur of Cambodia | PID:Pv611 | Fresh isolate fed to *An. dirus* mosquitoes |
| Biological sample (*Plasmodium vivax*) | Patient isolate | Institute Pasteur of Cambodia | PID:Pv623 | Fresh isolate fed to *An. dirus* mosquitoes |
| Biological sample (*Plasmodium vivax*) | Patient isolate | Institute Pasteur of Cambodia | PID:Pv624 | Fresh isolate fed to *An. dirus* mosquitoes |
| Biological sample (*Plasmodium vivax*) | Patient isolate | Institute Pasteur of Cambodia | PID:Pv635 | Fresh isolate fed to *An. dirus* mosquitoes |
| Biological sample (*Plasmodium vivax*) | Patient isolate | Institute Pasteur of Cambodia | PID:Pv640 | Fresh isolate fed to *An. dirus* mosquitoes |

*Appendix 1 Continued on next page*

*Appendix 1 Continued*

| Reagent type (species) or resource | Designation | Source or reference | Identifiers | Additional information |
|---|---|---|---|---|
| Biological sample (*Plasmodium vivax*) | Patient isolate | Institute Pasteur of Cambodia | PID:Pv644 | Fresh isolate fed to *An. dirus* mosquitoes |
| Biological sample (*Plasmodium vivax*) | Patient isolate | Institute Pasteur of Cambodia | PID:Pv708 | Fresh isolate fed to *An. dirus* mosquitoes |
| Biological sample (*Plasmodium vivax*) | Patient isolate | Institute Pasteur of Cambodia | PID:Pv836 | Fresh isolate fed to *An. dirus* mosquitoes |
| Biological sample (*Plasmodium vivax*) | Patient isolate | Institute Pasteur of Cambodia | PID:Pv838 | Fresh isolate fed to *An. dirus* mosquitoes |
| Biological sample (*Plasmodium vivax*) | Patient isolate | Institute Pasteur of Cambodia | PID:Pv846 | Fresh isolate fed to *An. dirus* mosquitoes |
| Biological sample (*Plasmodium vivax*) | Patient isolate | Institute Pasteur of Cambodia | PID:Pv847 | Fresh isolate fed to *An. dirus* mosquitoes |
| Biological sample (*Plasmodium vivax*) | Patient isolate | Institute Pasteur of Cambodia | PID:Pv849 | Fresh isolate fed to *An. dirus* mosquitoes |
| Biological sample (*Plasmodium vivax*) | Patient isolate | Institute Pasteur of Cambodia | PID:Pv893 | Fresh isolate fed to *An. dirus* mosquitoes |
| Biological sample (*Plasmodium vivax*) | Patient isolate | Institute Pasteur of Cambodia | PID:Pv922 | Fresh isolate fed to *An. dirus* mosquitoes |
| Biological sample (*Plasmodium vivax*) | Patient isolate | Institute Pasteur of Cambodia | PID:Pv923 | Fresh isolate fed to *An. dirus* mosquitoes |
| Biological sample (*Plasmodium vivax*) | Patient isolate | Institute Pasteur of Cambodia | PID:Pv950 | Fresh isolate fed to *An. dirus* mosquitoes |
| Biological sample (*Plasmodium vivax*) | Patient isolate | Institute Pasteur of Cambodia | PID:Pv951 | Fresh isolate fed to *An. dirus* mosquitoes |
| Biological sample (*Plasmodium vivax*) | Patient isolate | Institute Pasteur of Cambodia | PID:Pv952 | Fresh isolate fed to *An. dirus* mosquitoes |
| Biological sample (*Plasmodium vivax*) | Patient isolate | Institute Pasteur of Cambodia | PID:Pv959 | Fresh isolate fed to *An. dirus* mosquitoes |
| Biological sample (*Plasmodium vivax*) | Patient isolate | Institute Pasteur of Cambodia | PID:Pv1014 | Fresh isolate fed to *An. dirus* mosquitoes |
| Biological sample (*Plasmodium vivax*) | Patient isolate | Institute Pasteur of Cambodia | PID:Pv1020 | Fresh isolate fed to *An. dirus* mosquitoes |
| Biological sample (*Plasmodium vivax*) | Patient isolate | Institute Pasteur of Cambodia | PID:Pv1024 | Fresh isolate fed to *An. dirus* mosquitoes |
| Biological sample (*Plasmodium vivax*) | Patient isolate | Institute Pasteur of Cambodia | PID:IV21-075 | Fresh isolate fed to *An. dirus* mosquitoes |
| Biological sample (*Plasmodium vivax*) | Patient isolate | Institute Pasteur of Cambodia | PID:PQRC21-113 | Fresh isolate fed to *An. dirus* mosquitoes |
| Biological sample (*Plasmodium vivax*) | Patient isolate | Institute Pasteur of Cambodia | PID:PQRC21-135 | Fresh isolate fed to *An. dirus* mosquitoes |
| Biological sample (*Plasmodium vivax*) | Patient isolate | Mahidol Vivax Research Unit | PID:VTTY201 | Fresh isolate fed to *An. dirus* mosquitoes |
| Biological sample (*Plasmodium cynomolgi*) | M/B strain | PMID:31536608 | | Emory National Primate Research Center |
| Biological sample (*Plasmodium cynomolgi*) | Rossan strain | PMID:18788885 | | Emory National Primate Research Center |
| Biological sample (*Plasmodium cynomolgi*) | B strain | PMID:32660993 | | Armed Forces Research Institute of Medical Sciences |

*Appendix 1 Continued on next page*

*Appendix 1 Continued*

| Reagent type (species) or resource | Designation | Source or reference | Identifiers | Additional information |
|---|---|---|---|---|
| Biological sample (*Plasmodium cynomolgi*) | *P. berghei* ANKA strain GFP Luc$_{ama1\text{-}eef1a}$ (line 1052cl1) | PMID:36100902 | | University of Georgia |
| Biological sample (*Plasmodium falciparum*) | Dd2-HLH | BEI Resources | Cat#:MRA-156 | |
| Biological sample (*Plasmodium cynomolgi*) | DC | This paper | | University of Georgia |
| Biological sample (*Plasmodium falciparum*) | W2 | PMID:7729473 | | University of Georgia |
| Strain, strain background (*Macaca fuscata*, male) | Monkey, used for experimental animal infection | Emory National Primate Research Center | | Not genetically modified |
| Strain, strain background (*Macaca fuscata*, male) | Monkey, used for experimental animal infection | Emory National Primate Research Center | | Not genetically modified |
| Antibody | anti *P. vivax* Upregulated in Infectious Sporozoites 4 (r*Pv*UIS4) (recombinant mouse monoclonal) | PMID:30333026 | | IFA (1:10,000) |
| Antibody | anti-*Pv*MIF (rabbit polyclonal) | PMID:25800544 | | IFA (1:1000) |
| Antibody | anti-*Pc*HSP70 (rabbit polyclonal) | This paper | n/a | IFA (200 ng/ml) |
| Antibody | anti-*Pc*UIS4 (mouse monoclonal) | This paper | n/a | IFA (10 ng/ml) |
| Antibody | anti-Mouse IgG (H+L) Cross-Adsorbed Secondary Antibody, Alexa Fluor 488 (Goat monoclonal) | Thermo Fisher Scientific | Cat#: A-11001; RRID:AB_2534069 | IFA (1:1000) |
| Antibody | anti-Human IgG (H+L) Cross-Adsorbed Secondary Antibody, Alexa Fluor 488 (Goat monoclonal) | Thermo Fisher Scientific | Cat#:A11013; RRID:AB_2534080 | IFA (1:1000) |
| Antibody | anti-Rabbit IgG (H+L) Highly Cross-Adsorbed Secondary Antibody, Alexa Fluor 568 (Goat monoclonal) | Thermo Fisher Scientific | Cat#:A11036; RRID:AB_10563566 | IFA (1:1000) |
| Antibody | 5-Methylcytosine Recombinant Antibody (rabbit monoclonal) | Thermo Fisher Scientific | Cat#:MA5-24694: RRID:AB_2665309; Clone:RM231 | 10 µg/ml |
| Antibody | 5-Hydroxymethylcytosine Recombinant Antibody (rabbit monoclonal) | Thermo Fisher Scientific | Cat#:MA5-24695; RRID:AB_2665308; Clone:RM236 | 10 µg/ml |
| Antibody | anti-Rabbit IgG (H+L) Cross-Adsorbed Secondary Antibody, Texas Red (goat monoclonal) | Thermo Fisher Scientific | Cat#:T-2767; RRID:AB_2556776 | 10 µg/ml |
| Antibody | anti-Plasmodium GAPDH (mouse monoclonal) | European Malaria Reagent Repository | Cat#:13.3 | 100 ng/ml |
| Software, algorithm | Genedata Screener, Version 15.0.1-Standard | Genedata | | |
| Chemical compound, drug | Budralazine | Chemcruz | Cat3:sc-504334 | Batch D3019 |
| Chemical compound, drug | Cadralazine | Chemcruz | Cat#:sc-500641 | Batch B24217 |
| Chemical compound, drug | Hydralazine | Selleckchem | Cat#:S2562 | Batch S256202 |

*Appendix 1 Continued on next page*

*Appendix 1 Continued*

| Reagent type (species) or resource | Designation | Source or reference | Identifiers | Additional information |
|---|---|---|---|---|
| Chemical compound, drug | Dihydralazine | Calibr at Scripps | Code:CBR-001-571-820-4 | |
| Chemical compound, drug | Plasmocid | Calibr at Scripps | Code:CBR-001-572-110-5 | |
| Chemical compound, drug | MS-0735 | Calibr at Scripps | Code:CBR-001-572-134-3 | |
| Chemical compound, drug | Hydralazine | Calibr at Scripps | Code:CBR-001-572-134-3 | |
| Chemical compound, drug | Colforsin daropate | Calibr at Scripps | Code:CBR-001-586-408-1 | |
| Chemical compound, drug | PAN-811 | Calibr at Scripps | Code:CBR-001-586-749-9 | |
| Chemical compound, drug | Todralazine | Calibr at Scripps | Code:CBR-001-586-916-6 | |
| Chemical compound, drug | RGH-5526 | Calibr at Scripps | Code:CBR-001-587-032-3 | |
| Chemical compound, drug | Budralazine | Calibr at Scripps | Code:CBR-001-587-246-5 | |
| Chemical compound, drug | Dramedilol | Calibr at Scripps | Code:CBR-001-593-286-2 | |
| Chemical compound, drug | Endralazine | Calibr at Scripps | Code:CBR-001-597-262-0 | |
| Chemical compound, drug | Cadralazine | Calibr at Scripps | Code:CBR-001-624-776-0 | |
| Chemical compound, drug | Pildralazine | Calibr at Scripps | Code:CBR-001-635-378-9 | |
| Chemical compound, drug | Mopidralazine | Calibr at Scripps | Code:CBR-001-635-852-4 | |
| Chemical compound, drug | Rhodamine 123 | Calibr at Scripps | Code:CBR-050-127-020-8 | |
| Chemical compound, drug | Narasin | Calibr at Scripps | Code:CBR-050-127-705-0 | |
| Chemical compound, drug | Poziotinib | Calibr at Scripps | Code:CBR-001-574-260-6 | |
| Chemical compound, drug | Panobinostat | Targetmol | Cat#:T2383 | |
| Chemical compound, drug | Abexinostat | Targetmol | Cat#:T0431 | |
| Chemical compound, drug | Pracinostat | Targetmol | Cat#:T1890 | |
| Chemical compound, drug | Cyproheptadine | Targetmol | Cat#:T0174 | |
| Chemical compound, drug | Cerdulatinib | Targetmol | Cat#:T2487 | |
| Chemical compound, drug | MI2 | Targetmol | Cat#:T2649 | |
| Chemical compound, drug | Raddeanin A | Targetmol | Cat#:T3878 | |
| Chemical compound, drug | CCT241736 | Targetmol | Cat#:T4428 | |

*Appendix 1 Continued on next page*

*Appendix 1 Continued*

| Reagent type (species) or resource | Designation | Source or reference | Identifiers | Additional information |
|---|---|---|---|---|
| Chemical compound, drug | 666-15 | Targetmol | Cat#:T5318 | |
| Chemical compound, drug | Givinostat | Targetmol | Cat#:T6279 | |
| Chemical compound, drug | AR42 | Targetmol | Cat#:T6392 | |
| Chemical compound, drug | MMV019721 | Medicines for Malaria Venture | Code:MMV019721 | Batch:MMV019721-08, MMV019721-10 |
| Chemical compound, drug | MMV084978 | Medicines for Malaria Venture | Code:MMV084978 | Batch:MMV084978-04, MMV084978-05 |
| Chemical compound, drug | 5-Azacytidine | Cyamen Chem | Cat#:11164 | |
| Chemical compound, drug | 1-Aminobenzotriazole | Cyamen Chem | Cat#:15252 | |
| Commercial assay or kit | Cell-Titer Glo | Promega | Cat#:G7573 | |
| Commercial assay or kit | EpiQuik DNA Methyltransferase (DNMT) Activity/Inhibition Assay Kit | EpiGentek | Cat#:P-3010 | |
| Commercial assay or kit | Epitect fast bisulfite conversion kit | QIAGEN | Cat#:59824 | |
| Commercial assay or kit | CYP3A4 luciferin-IPA kit | Promega | Cat#:V9001 | Used Lytic protocol |

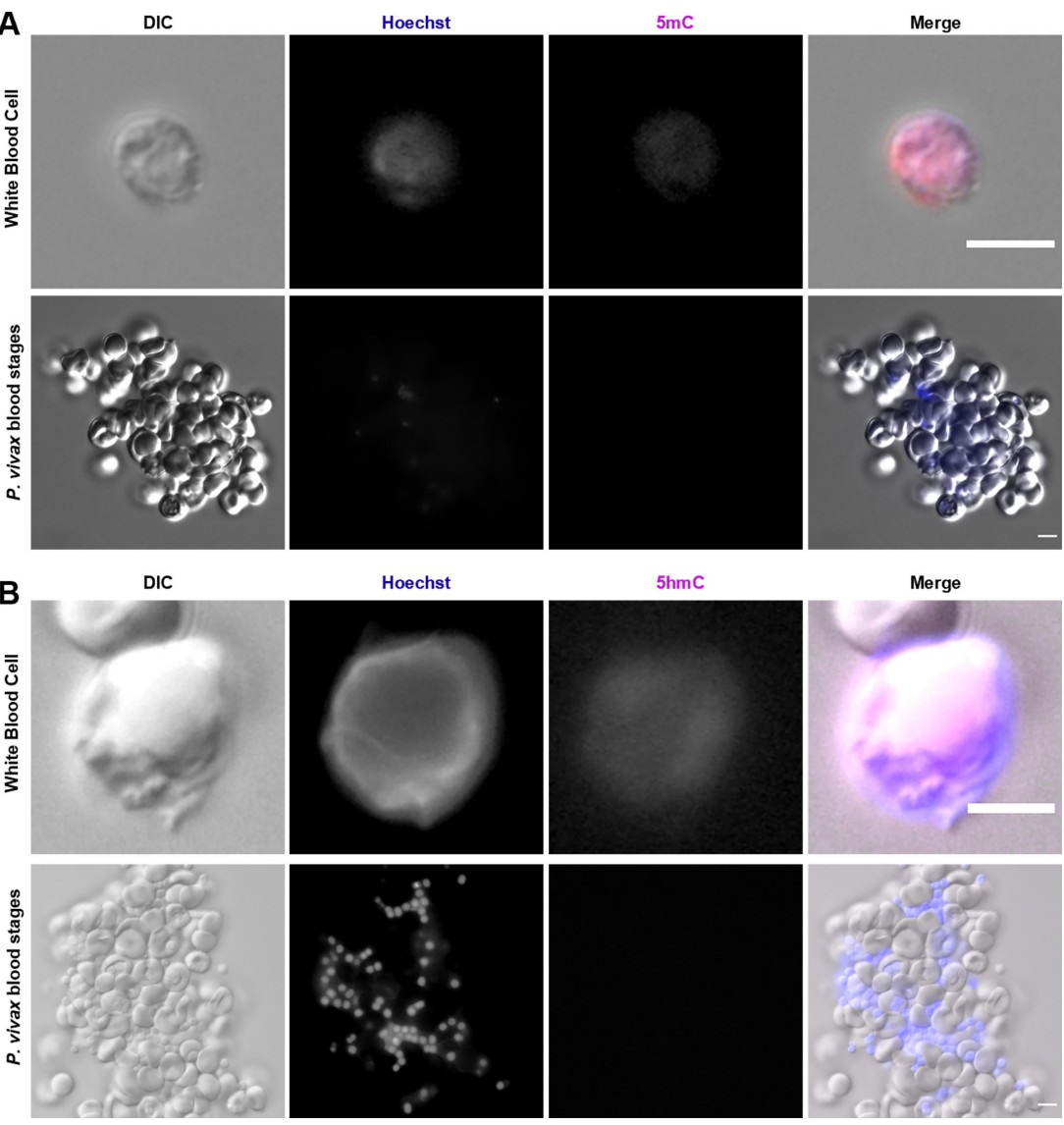

**Appendix 1—figure 1.** Cytosine modification in *P. vivax* blood stages. (**A**) *P. vivax* blood stages from patient isolates appeared negative when stained with 5mC. A white blood cell positive for 5mC serves as a stain control. (**B**) *P. vivax* blood stages from patient isolates appeared negative when stained with 5hmC. A white blood cell positive for 5hmC serves as a stain control. Bars represent 10 μm.

