## [Editor Report · eLife assessment]

This paper reports a large drug repurposing screen based on an in vitro culture platform to identify compounds that can kill Plasmodium hypnozoites. This **valuable** work adds to the current repertoire of anti-hypnozoites agents and uncovers targetable epigenetic pathways to enhance our understanding of this mysterious stage of the Plasmodium life cycle. The data presented here are based on **solid** methodology and represent a starting point for further investigation of epigenetic inhibitors to treat P. vivax infection. This paper will be of interest to Plasmodium researchers and more broadly to readers in the fields of host-pathogen interactions and drug development.

---

## [Referee Report · Reviewer #1 (Public Review)]

Summary:

Plasmodium vivax can persist in the liver of infected individuals in the form of dormant hypnozoites, which cause malaria relapses and are resistant to most current antimalarial drugs. This highlights the need to develop new drugs active against hypnozoites that could be used for radical cure. Here, the authors capitalize on an in vitro culture system based on primary human hepatocytes infected with P. vivax sporozoites to screen libraries of repurposed molecules and compounds acting on epigenetic pathways. They identified a number of hits, including hydrazinophthalazine analogs. They propose that some of these compounds may act on epigenetic pathways potentially involved in parasite quiescence. To provide some support to this hypothesis, they document DNA methylation of parasite DNA based on 5-methylcytosine immunostaining, mass spectrometry, and bisulfite sequencing.

Strengths:

-The drug screen itself represents a huge amount of work and, given the complexity of the experimental model, is a tour de force.

-The screening was performed in two different laboratories, with a third laboratory being involved in the confirmation of some of the hits, providing strong support that the results were reproducible.

-The screening of repurposing libraries is highly relevant to accelerate the development of new radical cure strategies.

Weaknesses:

-The manuscript is composed of two main parts, the drug screening itself and the description of DNA methylation in Plasmodium pre-erythrocytic stages. Unfortunately, these two parts are loosely connected. First, there is no evidence that the identified hits kill hypnozoites via epigenetic mechanisms. The hit compounds almost all act on schizonts in addition to hypnozoites, therefore it is unlikely that they target quiescence-specific pathways. At least one compound, colforsin, seems to selectively act on hypnozoites, but this observation still requires confirmation. Second, while the description of DNA methylation is per se interesting, its role in quiescence is not directly addressed here. Again, this is clearly not a specific feature of hypnozoites as it is also observed in *P. vivax* and *P. cynomolgi* hepatic schizonts and in *P. falciparum* blood stages. Therefore, the link between DNA methylation and hypnozoite formation is unclear. In addition, DNA methylation in sporozoites may not reflect epigenetic regulation occurring in the subsequent liver stages.

-The mode of action of the hit compounds remains unknown. In particular, it is not clear whether the drugs act on the parasite or on the host cell. Merely counting host cell nuclei to evaluate the toxicity of the compounds is probably acceptable for the screen but may not be sufficient to rule out an effect on the host cell. A more thorough characterization of the toxicity of the selected hit compounds is required.

-There is no convincing explanation for the differences observed between *P. vivax* and *P. cynomolgi*. The authors question the relevance of the simian model but the discrepancy could also be due to the *P. vivax* in vitro platform they used.

-Many experiments were performed only once, not only during the screen (where most compounds were apparently tested in a single well) but also in other experiments. The quality of the data would be increased with more replication.

-While the extended assay (12 days versus 8 days) represents an improvement of the screen, the relevance of adding inhibitors of core cytochrome activity is less clear, as under these conditions the culture system deviates from physiological conditions.

---

## [Referee Report · Reviewer #2 (Public Review)]

Summary:

In this manuscript, inhibitors of the P. vivax liver stages are identified from the Repurposing, Focused Rescue, and Accelerated Medchem (ReFRAME) library as well as a 773-member collection of epigenetic inhibitors. This study led to the discovery that epigenetics pathway inhibitors are selectively active against P. vivax and P. cynomolgi hypnozoites. Several inhibitors of histone post-translational modifications were found among the hits and genomic DNA methylation mapping revealed the modification on most genes. Experiments were completed to show that the level of methylation upstream of the gene (promoter or first exon) may impact gene expression. With the limited number of small molecules that act against hypnozoites, this work is critically important for future drug leads. Additionally, the authors gleaned biological insights from their molecules to advance the current understanding of essential molecular processes during this elusive parasite stage.

Strengths:

-This is a tremendously impactful study that assesses molecules for the ability to inhibit Plasmodium hypnozoites. The comparison of various species is especially relevant for probing biological processes and advancing drug leads.

-The SI is wonderfully organized and includes relevant data/details. These results will inspire numerous studies beyond the current work.

---

## [Referee Report · Reviewer #3 (Public Review)]

Although this work represents a massive screening effort to find new drugs targeting P. vivax hypnozoites, the authors should balance their statement that they identified targetable epigenetic pathways in hypnozoites.

• They should emphasize the potential role of the host cell in the presentation of the results and the discussion, as it is known that other pathogens modify the epigenome of the host cell (i.e. toxoplasma, HIV) to prevent cell division. Also, hydrazinophtalazines target multiple pathways (notably modulation of calcium flux) and have been shown to inhibit DNA-methyl transferase 1 which is lacking in Plasmodium.

• In a drug repurposing approach, the parasite target might also be different than the human target.

• The authors state that host-cell apoptotic pathways are downregulated in P. vivax infected cells (p. 5 line 162). Maybe the HDAC inhibitors and DNA-methyltransferase inhibitors are reactivating these pathways, leading to parasite death, rather than targeting parasites directly.

It would make the interpretation of the results easier if the authors used EC50 in µM rather than pEC50 in tables and main text. It is easy to calculate when it is a single-digit number but more complicated with multiple digits.

Authors mention hypnozoite-specific effects but in most cases, compounds are as potent on hypnozoite and schizonts. They should rather use "liver stage specific" to refer to increased activity against hypnozoites and schizonts compared to the host cell. The same comment applies to line 351 when referring to MMV019721. Following the same idea, it is a bit far-fetched to call MMV019721 "specific" when the highest concentration tested for cytotoxicity is less than twice the EC50 obtained against hypnozoites and schizonts.

Page 5 lines 187-189, the authors state "...hydrazinophtalazines were inactive when tested against P. berghei liver schizonts and *P. falciparum* asexual blood stages, suggesting that hypnozoite quiescence may be biologically distinct from developing schizonts". The data provided in Figure 1B show that these hydrazinophtalazines are as potent in P. vivax schizonts than in P. vivax hypnozoites, so the distinct activity seems to be Plasmodium species specific and/or host-cell specific (primary human hepatocytes rather than cell lines for P. berghei) rather than hypnozoite vs schizont specific.

Why choose to focus on cadralazine if abandoned due to side effects? Also, why test the pharmacokinetics in monkeys? As it was a marketed drug, were no data available in humans?

In the counterscreen mentioned on page 6, the authors should mention that the activity of poziotinib in P. berghei and P. cynomolgi is equivalent to cell toxicity, so likely not due to parasite specificity.

To improve the clarity and flow of the manuscript, could the authors make a recapitulative table/figure for all the data obtained for poziotinib and hydrazinophtalazines in the different assays (8-days vs 12-days) and laboratory settings rather than separate tables in main and supplementary figures. Maybe also reorder the results section notably moving the 12-day assay before the DNA methylation part.

The isobologram plot shows an additive effect rather than a synergistic effect between cadralazine and 5-azacytidine, please modify the paragraph title accordingly. Please put the same axis scale for both fractional EC50 in the isobologram graph (Figure 2A).

Concerning the immunofluorescence detection of 5mC and 5hmC, the authors should be careful with their conclusions. The Hoechst signal of the parasites is indistinguishable because of the high signal given by the hepatocyte nuclei. The signal obtained with the anti-5hmC in hepatocyte nuclei is higher than with the anti-5mC, thus if a low signal is obtained in hypnozoites and schizonts, it might be difficult to dissociate from the background. In blood stages (Figure S18), the best to obtain a good signal is to lyse the red blood cell using saponin, before fixation and HCl treatment.

To conclude that 5mC marks are the predominate DNA methylation mark in both *P. falciparum* and *P. vivax*, authors should also mention that they compare different stages of the life cycle, that might have different methylation levels.

Also, the authors conclude that "[...] 5mC is present at low level in P. vivax and P. cynomolgi sporozoites and could control liver stage development and hypnozoite quiescence". Based on the data shown here, nothing, except presence the of 5mC marks, supports that DNA methylation could be implicated in liver stage development or hypnozoite quiescence.

How many DNA-methyltransferase inhibitors were present in the epigenetic library? Out of those, none were identified as hits, maybe the hydrazinophtalazines effect is not linked to DNMT inhibition but another target pathway of these molecules like calcium transport?

The authors state (line 344): "These results corroborate our hypothesis that epigenetic pathways regulate hypnozoites". This conclusion should be changed to "[...] that epigenetic pathways are involved in P. vivax liver stage survival" because:

• The epigenetic inhibitors described here are as active on hypnozoite than liver schizonts.

• Again, we cannot rule out that the host cell plays a role in this effect and that the compound may not act directly on the parasite.

The same comment applies to the quote in lines 394 to 396. There is no proof in the results presented here that DNA methylation plays any role in the effect of hydrazinophtalazines in the anti-plasmodial activity obtained in the assay.

---

## [Author Response]

**Public Reviews:**

**Reviewer #1 (Public Review):**
Summary:Plasmodium vivax can persist in the liver of infected individuals in the form of dormant hypnozoites, which cause malaria relapses and are resistant to most current antimalarial drugs. This highlights the need to develop new drugs active against hypnozoites that could be used for radical cure. Here, the authors capitalize on an in vitro culture system based on primary human hepatocytes infected with P. vivax sporozoites to screen libraries of repurposed molecules and compounds acting on epigenetic pathways. They identified a number of hits, including hydrazinophthalazine analogs. They propose that some of these compounds may act on epigenetic pathways potentially involved in parasite quiescence. To provide some support to this hypothesis, they document DNA methylation of parasite DNA based on 5-methylcytosine immunostaining, mass spectrometry, and bisulfite sequencing.Strengths:-The drug screen itself represents a huge amount of work and, given the complexity of the experimental model, is a tour de force.-The screening was performed in two different laboratories, with a third laboratory being involved in the confirmation of some of the hits, providing strong support that the results were reproducible.-The screening of repurposing libraries is highly relevant to accelerate the development of new radical cure strategies.

We thank the reviewer for pointing out the strengths of our report.

Weaknesses:The manuscript is composed of two main parts, the drug screening itself and the description of DNA methylation in Plasmodium pre-erythrocytic stages. Unfortunately, these two parts are loosely connected. First, there is no evidence that the identified hits kill hypnozoites via epigenetic mechanisms. The hit compounds almost all act on schizonts in addition to hypnozoites, therefore it is unlikely that they target quiescence-specific pathways. At least one compound, colforsin, seems to selectively act on hypnozoites, but this observation still requires confirmation. Second, while the description of DNA methylation is per se interesting, its role in quiescence is not directly addressed here. Again, this is clearly not a specific feature of hypnozoites as it is also observed in P. vivax and P. cynomolgi hepatic schizonts and in *P. falciparum* blood stages. Therefore, the link between DNA methylation and hypnozoite formation is unclear. In addition, DNA methylation in sporozoites may not reflect epigenetic regulation occurring in the subsequent liver stages.

We agree our report lacks direct evidence that hydrazinophthalazines are interacting with parasite epigenetic mechanisms. We spent significant resources attempting several novel approaches to establish a direct connection, but technological advances are needed to enable such studies, which we mention in the introduction and discussion. We disagree that schizonticidal activity automatically excludes the possibility a hypnozonticidal hit is acting on quiescence-specific pathways because both hypnozoites and schizonts are under epigenetic control and these pathways are likely performing different functions in different stages. Also important is the use of the word ‘specific’ as this term could be used to indicate parasite versus host (a drug that clears a parasite infection with a safety margin), parasite-directed effect versus host-directed effect (a drug acting via an agonistic or antagonistic effect on parasite or host pathway(s), but leading to parasite death in either case), hypnozoite versus schizont, or P. vivax versus other Plasmodium species. We were careful to indicate the usage of ‘specific’ throughout the text. Given the almost-nonexistent hit rate when screening diverse small molecule libraries screening against P. vivax hypnozoites, and remarkable increase in hits when screening epigenetic inhibitors as described in this report, our data suggests epigenetic pathways are important to the regulation of hypnozoite dormancy in addition to regulation of other parasite stages, but those effects are outside the scope of this report.

-The mode of action of the hit compounds remains unknown. In particular, it is not clear whether the drugs act on the parasite or on the host cell. Merely counting host cell nuclei to evaluate the toxicity of the compounds is probably acceptable for the screen but may not be sufficient to rule out an effect on the host cell. A more thorough characterization of the toxicity of the selected hit compounds is required.

We agree, and mention in the results and discussion, that the effect could be mediated through host pathways. This is not unlike the 8-aminoquinolones, which are activated by host cytochromes and kill via ROS, which is a nonspecific mechanism (that is, the compound is not directly interacting with a parasite target) leading to a parasite-specific effect (the parasite cannot tolerate the ROS produced, but the host can). During screening, it is generally the case that detecting hits with direct effects on the target organism are more desirable, so hits are counterscreened for general cytotoxicity. In this report, we show an effect on the parasite in direct comparison to the effect on host primary hepatocytes in the P. vivax assay itself, and follow up on hits with general counterscreens using two mammalian cell lines using CellTiter Glo, which does not rely on nuclei counts. Some compounds did show general cytotoxic effects, but with selectivity (more potency) against P. vivax liver stages, while other hits like the hydrazinophthalazines did not show an effect against primary hepatocytes and show only weak toxicity against mammalian cells at the highest dose tested. Further studies are needed to determine if the effect is indeed host- or parasite-directed and, if hydrazinophthalazines are to be developed into marketed antimalarials, extensive safety testing would be part of the development process.

-There is no convincing explanation for the differences observed between P. vivax and P. cynomolgi. The authors question the relevance of the simian model but the discrepancy could also be due to the P. vivax in vitro platform they used.

Fully characterizing the chemo-sensitivity of P. vivax and P. cynomolgi liver stages is outside the scope of this report. Rather, we report tool compounds which could be used in future studies to further characterize these sister species. We also make the point that P. cynomolgi is the gold standard for in vivo antirelapse activity, but it is still a model species, not a target species, and so few experimental hypnozonticidal compounds have been reported that the predictive value of P. cynomolgi is not fully understood. We found that several of our hits were species-specific using our in vitro platforms, thus future studies are needed to ensure this predictive value.

-Many experiments were performed only once, not only during the screen (where most compounds were apparently tested in a single well) but also in other experiments. The quality of the data would be increased with more replication.

Due to their size, compound library screens are typically performed once, with confirmation in dose-response assays, which were repeated several times. Rhesus PK studies was performed once on three animals, which is typical. All other studies were performed at least twice and most were performed three times or more. We provide a data table showing readers the source material for all replication as well as other source data tables showing the raw data for dose-response and other assays.

-While the extended assay (12 days versus 8 days) represents an improvement of the screen, the relevance of adding inhibitors of core cytochrome activity is less clear, as under these conditions the culture system deviates from physiological conditions.

We agree that cytochrome inhibitors render the platform less physiologically relevant, but the goal of screening is to detect hits which could be improved upon using medicinal chemistry, including metabolic stability. Metabolic stability is better assessed using standard assays such as liver microsomes, thus our goal was to characterize the effects of test compounds on the parasite without the confounding effect of hepatic metabolism.

**Reviewer #2 (Public Review):**
Summary:In this manuscript, inhibitors of the P. vivax liver stages are identified from the Repurposing, Focused Rescue, and Accelerated Medchem (ReFRAME) library as well as a 773-member collection of epigenetic inhibitors. This study led to the discovery that epigenetics pathway inhibitors are selectively active against P. vivax and P. cynomolgi hypnozoites. Several inhibitors of histone post-translational modifications were found among the hits and genomic DNA methylation mapping revealed the modification on most genes. Experiments were completed to show that the level of methylation upstream of the gene (promoter or first exon) may impact gene expression. With the limited number of small molecules that act against hypnozoites, this work is critically important for future drug leads. Additionally, the authors gleaned biological insights from their molecules to advance the current understanding of essential molecular processes during this elusive parasite stage.Strengths:-This is a tremendously impactful study that assesses molecules for the ability to inhibit Plasmodium hypnozoites. The comparison of various species is especially relevant for probing biological processes and advancing drug leads.-The SI is wonderfully organized and includes relevant data/details. These results will inspire numerous studies beyond the current work.

We thank the reviewer for pointing out the strengths of our report.

**Reviewer #3 (Public Review):**
Although this work represents a massive screening effort to find new drugs targeting P. vivax hypnozoites, the authors should balance their statement that they identified targetable epigenetic pathways in hypnozoites.-They should emphasize the potential role of the host cell in the presentation of the results and the discussion, as it is known that other pathogens modify the epigenome of the host cell (i.e. toxoplasma, HIV) to prevent cell division. Also, hydrazinophtalazines target multiple pathways (notably modulation of calcium flux) and have been shown to inhibit DNA-methyl transferase 1 which is lacking in Plasmodium.-In a drug repurposing approach, the parasite target might also be different than the human target.-The authors state that host-cell apoptotic pathways are downregulated in P. vivax infected cells (p. 5 line 162). Maybe the HDAC inhibitors and DNA-methyltransferase inhibitors are reactivating these pathways, leading to parasite death, rather than targeting parasites directly.

We agree caution must be taken as we did not directly confirm the mechanism of our hits. Many follow up studies will be needed to do so. We do point out in the discussion that the mechanism of hits could be host-directed. We agree with the notion that some of these hits could be affecting parasitized host cell pathways, which lead to death of the parasitized cell, with the parasite being collateral damage, yet such a mechanism could lead to a safe and effective novel antimalarial.

It would make the interpretation of the results easier if the authors used EC50 in µM rather than pEC50 in tables and main text. It is easy to calculate when it is a single-digit number but more complicated with multiple digits.

We apologize for the atypical presentation of potency data. However, there is growing concern in drug discovery when Standard Deviation is applied to Potency data because Standard Deviation is a linear calculation and Potency is a log effect, making the math incompatible. We understand thousands of papers are reported every year using this mathematically incorrect method, making our presentation of these data less familiar. However, we define pEC50 in its use in the text and table legends and hope to increase its use in the broader scientific community.

Authors mention hypnozoite-specific effects but in most cases, compounds are as potent on hypnozoite and schizonts. They should rather use "liver stage specific" to refer to increased activity against hypnozoites and schizonts compared to the host cell. The same comment applies to line 351 when referring to MMV019721. Following the same idea, it is a bit far-fetched to call MMV019721 "specific" when the highest concentration tested for cytotoxicity is less than twice the EC50 obtained against hypnozoites and schizonts.

We have reviewed and revised statements in the manuscript to ensure the effect we are describing is accurate in terms of parasite versus parasite form.

Page 5 lines 187-189, the authors state "...hydrazinophtalazines were inactive when tested against P. berghei liver schizonts and *P. falciparum* asexual blood stages, suggesting that hypnozoite quiescence may be biologically distinct from developing schizonts". The data provided in Figure 1B show that these hydrazinophtalazines are as potent in P. vivax schizonts than in P. vivax hypnozoites, so the distinct activity seems to be Plasmodium species specific and/or host-cell specific (primary human hepatocytes rather than cell lines for P. berghei) rather than hypnozoite vs schizont specific.

We agree the effect of hydrazinophtalazine could be more species specific than stage specific, but the context of our comment has to do with current methods in antimalarial discovery and development. Given the biological uniqueness of the various Plasmodium species and stages, any hypnozonticidal hit may or may not have pan-species or pan-stage activity; our goal was to characterize this. Regardless of the mechanism, we found it interesting that the hydrazinophtalazines kill P. vivax hypnozoites, but not P. cynomolgi hypnozoites nor other species and stages used in antimalarial drug development. This result makes the point that hypnozoite-focused assays may be required to detect and develop hypnozonticidal hits, regardless of what other species or stages they may or may not act on.

Why choose to focus on cadralazine if abandoned due to side effects? Also, why test the pharmacokinetics in monkeys? As it was a marketed drug, were no data available in humans?Cadralazine was found more potent than hydralazine and PK data was available from humans, thus dose prediction calculations showed an efficacious dose was more achievable with cadralazine than hydralazine. Side effects are often dependent on dose and regimen, which are very likely to be much different for treating malaria versus hypertension. Thus, the potential side effects of cadralazine if it was to be used as an antimalarial are simply unknown and are not disqualifying at this step. The PK study was done in Rhesus macaques so we could calculate the dose needed to achieve coverage of EC90 during a planned follow up in a Rhesus-P. cynomolgi relapse model. However, this planned in vivo efficacy study was not justified once we concurrently discovered cadralazine was inactive on P. cynomolgi in vitro.In the counterscreen mentioned on page 6, the authors should mention that the activity of poziotinib in P. berghei and P. cynomolgi is equivalent to cell toxicity, so likely not due to parasite specificity.

Poziotinib shows activity against mammalian cell lines but not against the primary hepatocyte cultures supporting dose-response assays against P. vivax liver forms, which do not replicate. Thus, poziotinib appears selective in the liver stage assay but also may have a much more potent effect in continuously replicating cell lines.

To improve the clarity and flow of the manuscript, could the authors make a recapitulative table/figure for all the data obtained for poziotinib and hydrazinophtalazines in the different assays (8-days vs 12-days) and laboratory settings rather than separate tables in main and supplementary figures. Maybe also reorder the results section notably moving the 12-day assay before the DNA methylation part.

We apologize for the large amount of data presented but believe we are presenting it in the clearest way possible. All raw data is available if readers wish to re-analyze or re-organize our findings.

The isobologram plot shows an additive effect rather than a synergistic effect between cadralazine and 5-azacytidine, please modify the paragraph title accordingly. Please put the same axis scale for both fractional EC50 in the isobologram graph (Figure 2A).

The isobologram shows the effect approaching synergy at some combinations. The isobologram was rendered using standard methods. The raw data is available if readers wish to re-analyze it.

Concerning the immunofluorescence detection of 5mC and 5hmC, the authors should be careful with their conclusions. The Hoechst signal of the parasites is indistinguishable because of the high signal given by the hepatocyte nuclei. The signal obtained with the anti-5hmC in hepatocyte nuclei is higher than with the anti-5mC, thus if a low signal is obtained in hypnozoites and schizonts, it might be difficult to dissociate from the background. In blood stages (Figure S18), the best to obtain a good signal is to lyse the red blood cell using saponin, before fixation and HCl treatment.

We spent many hours using high resolution imaging of hundreds of parasites trying to detect clear 5hmC signal in both hypnozoites and schizonts but never saw a clearly positive signal. Indeed, the host signal can be confounding, thus we felt the most clear and unbiased way to quantify and present these data was using HCI. We appreciate the suggestion to lyse cells first for detecting in the blood stage.

To conclude that 5mC marks are the predominate DNA methylation mark in both *P. falciparum* and *P. vivax*, authors should also mention that they compare different stages of the life cycle, that might have different methylation levels.

We do mention at the start of this section our reasoning that quantifying marks in sporozoites was technically achievable, but not in a mixed culture of parasites and hepatocytes. We agree they could have different marks at these different stages.

Also, the authors conclude that "[...] 5mC is present at low level in P. vivax and P. cynomolgi sporozoites and could control liver stage development and hypnozoite quiescence". Based on the data shown here, nothing, except presence the of 5mC marks, supports that DNA methylation could be implicated in liver stage development or hypnozoite quiescence.

We clearly show sporozoite and liver stage DNA is methylated, which implicates this fundamental cell function exists in P. vivax liver stages, and that compounds with characterized activity against DNMT are active on liver stages. We acknowledge we were unable to show a direct effect and use the qualifier ‘could’ for this very reason.

How many DNA-methyltransferase inhibitors were present in the epigenetic library? Out of those, none were identified as hits, maybe the hydrazinophtalazines effect is not linked to DNMT inhibition but another target pathway of these molecules like calcium transport?

We supply the complete list of inhibitors in the epigenetic library as a supplemental file, the library contained 773 compounds. Hydrazinophtalazines were not included in the library, but several other DNA methyltransferase inhibitors were inactive. It is possible that hydrazinophtalazine activity is linked to other mechanisms but the inactivity of other DNMT inhibitors does not preclude the possibility hydrazinophtalazines are acting through DNMT.

The authors state (line 344): "These results corroborate our hypothesis that epigenetic pathways regulate hypnozoites". This conclusion should be changed to "[...] that epigenetic pathways are involved in P. vivax liver stage survival" because:-The epigenetic inhibitors described here are as active on hypnozoite than liver schizonts.-Again, we cannot rule out that the host cell plays a role in this effect and that the compound may not act directly on the parasite.The same comment applies to the quote in lines 394 to 396. There is no proof in the results presented here that DNA methylation plays any role in the effect of hydrazinophtalazines in the anti-plasmodial activity obtained in the assay.

We maintain that we use words throughout the text that express uncertainty about the mechanisms involved. It is important to point out that, prior to this paper, the number of hypnozonticidal hits was incredibly low and this field is just emerging. The fundamental role of epigenetic mechanisms is regulation of gene expression. Finding several hypnozonticial hits when screening epigenetic libraries implies epigenetic pathways are important for hypnozoite survival. We intentionally do not specify exact mechanisms or if they are host or parasite pathways. Host-parasite interactions in the liver stage are incredibly difficult to resolve and are outside the scope of this report. Furthermore, this statement is not exclusive to schizonts, but since screens of diversity sets against schizonts result in a much higher hit rate, the focus of this comment is unearthing rare hypnozonticidal hits.